# Influence of Vineyard Location, Cluster Thinning and Spontaneous Alcoholic Fermentation on Wine Composition

**DOI:** 10.3390/foods14071101

**Published:** 2025-03-22

**Authors:** Franc Čuš, Anastazija Jež Krebelj, Mateja Potisek

**Affiliations:** Department of Fruit Growing, Viticulture, and Oenology, Agricultural Institute of Slovenia, Hacquetova ulica 17, 1000 Ljubljana, Slovenia; anastazija.jezkrebelj@kis.si (A.J.K.); mateja.potisek@kis.si (M.P.)

**Keywords:** terroir, crop load, spontaneous alcoholic fermentation, non-*Saccharomyces* yeasts

## Abstract

The influence of the vineyard location, the yield per vine and the type of alcoholic fermentation on the composition of Merlot wine from two consecutive vintages was investigated in a simultaneous experiment. Grapes from two locations and two crop loads per vine, from controlled and thinned vines, were vinified. At the same time, grapes from control vines were vinified with inoculated and spontaneous alcoholic fermentation. Comparisons of the wine composition were made using a targeted metabolomic approach, microbiological analysis and sensory evaluation. It has been confirmed that the composition of Merlot wine is essentially determined by the location of the vineyard. The analytical marker used to distinguish the two locations was the content of 3-mercaptohexan-1-ol (significantly higher in location B with 38–130%). It has also been shown that the type of alcoholic fermentation has a greater influence on the composition of the wine than the crop load. The analytical marker used for the cluster thinning was the pH of the wine, which increased significantly by 0.03 to 0.08 units with the lower crop load, and for the type of alcoholic fermentation, the concentration of 2-phenethyl acetate, which relates to the sum of acetates and 2-phenylethanol, which increased significantly by 58–299%, 54–218%, and 24–46% in the spontaneously fermented wines. Both the location of the vineyard and spontaneous alcoholic fermentation influenced the significant differences in the sensory characteristics of the wine, while cluster thinning had no such influence. The other influences of the two technical factors on the wine composition depended on the location of the vineyard and the vintage. It can also be concluded that spontaneous alcoholic fermentation reduced the influence of the vintage on the wine composition, while the opposite was the case with cluster thinning.

## 1. Introduction

The quality and style of the wine are primarily determined by the place where the vine grows. The most important factors at each location are the soil and the climatic conditions [1,2]. Many variables, such as the total soluble solids content, total titratable acidity, nitrogen content, and balance of phenolic compounds in grapes and wines, are therefore influenced by genetic (varietal) and environmental factors, known as terroir, and by the vintage [3]. Merlot grown in cool climates or with low sun exposure may be green due to the presence of 2-isobutyl-3-methoxypyrazine (IBMP), although this is rare in Merlot as it tends to ripen early. Merlot grown in temperate climates expresses fruity aromas and develops a complex ageing bouquet after a few years of bottle ageing. These positive characteristics are caused by a wide range of compounds, including substituted esters, varietal thiols (especially 3-mercaptohexan-1-ol (3MH)) and dimethyl sulphide (DMS) [2].

It has long been known through sensory evaluation that the taste of a wine from a particular variety can be associated with its origin and that wine typicity in relation to terroir is also largely shaped by aroma compounds [2]. However, the comparison between the sensory representations of wine as a concept and the descriptive methods revealed a gap between the conceptual and perceptual representation of typicality. On the one hand, the conceptual representation was consistent with soil as the first factor influencing typicity. On the other hand, the perceptual representation lacked consensus and emphasised the predominance of technical factors, particularly oenological factors, over environmental factors [4]. The management of microbial activity during winemaking is certainly one of the most important technical factors, as many wine aroma compounds are either undetectable or present as odourless bound precursors in the grapes formed during the alcoholic fermentation process. The microbiota of grapes and wine exhibits regionally defined patterns related to the vineyard and climatic conditions [5,6], but for a long time, it was not entirely clear to what extent these microbial patterns were related to the chemical composition (volatile compounds—VOCs) of the wine [6,7]. Recent studies provided new insights into the metabolic pathways of key flavour compounds formed by indigenous microorganisms during spontaneous alcoholic fermentation [8,9]. Traditional winemaking practices either encourage or rely entirely on fermentations with non-inoculated microorganisms. This practice is seen by its adherents as enhancing regional typicity [7] and could differentiate wine estates [9]. The microbiota and metabolites of grape and wine differ geographically, with several layers of fungal and bacteria microbiota identified that correlate with wine aroma profiles [6,8,9], and it has been shown that characteristic *S. cerevisiae* exert the strongest influences on wine quality and style at small geographic scales [10]. However, it was pointed out that experimental research is still needed to investigate the contribution of this assembly of microorganisms to the final product and to support or refute the concept of microbial terroir [11]. In uninoculated alcoholic fermentations (spontaneous alcoholic fermentations), there is a progressive growth pattern of indigenous yeasts, with the final stages invariably dominated by the Crabtree-positive, alcohol-tolerant strains of *S. cerevisiae* [12]. Of the indigenous yeasts other than *Saccharomyces*, *Hanseniaspora* spp. and *Starmerella*/*Candida* spp. are most prevalent in the early stages of many spontaneous alcoholic fermentations [8,9,13,14,15,16]. The *Hanseniaspora* spp. most commonly found on grapes and most associated with alcoholic fermentation are *H. vineae* [17], *H. osmophila* [18] and *H. uvarum* [19]. However, their effects on wine aroma appear to be a strain-specific trait, as only certain *Hanseniaspora* strains result in wines with increased levels of acetate esters such as 2-phenyl acetate and ethyl acetate, which are associated with fruity and floral aroma descriptions of wines [18,20]. The production of other secondary metabolites, including glycerol, acetaldehyde, ethyl acetate, and hydrogen sulphide, also differs among strains [21,22]. *Starmerella*/*Candida* spp. (*Candida zemplinina*/*Candida stellata*) strains, on the other hand, strongly influence the analytical profile of the wines, with a consistent increase in the levels of glycerol, succinic acid, terpenols, and lactones, and a decrease in the levels of acetic acid, aldehydes, 3-mercaptohexan-1-ol (3MH), 3-mercaptohexyl acetate (3MHA), and acetate esters compared to the values for *S. cerevisiae* fermentation, but the result depends on whether pure or mixed cultures are used, on the yeast strain, and on the grapevine variety [13,23,24,25,26]. The use of non-*Saccharomyces* strains can also intensify the colour of red wines after fermentation through the formation of large amounts of stable pyranoanthocyanin pigments [27], and *S. cerevisiae* strains also differ in their influence on the formation of stable pigments [28].

However, an important management technique in the vineyard is yield regulation, which influences the ratio between the leaf area and the fruit mass of the vine and has a considerable influence on the composition of the fruit and the quality of the wine [29,30]. To improve this ratio, winegrowers often use cluster thinning (CT), a management method in which clusters are selectively removed from the vines. There is still no consensus on the optimal timing and intensity of thinning to improve fruit composition [31]. Later authors summarised the results from the literature in a meta-analysis and reported, firstly, that the timing of thinning had little effect on fruit composition, giving growers more flexibility in applying this practise. Secondly, the severity of CT was important for improving fruit composition (TSS and pH), with only the moderate range (36–55%) being effective. The authors concluded that fruit composition is influenced more by the severity of CT than by the timing. In this meta-analysis, CT was also found to affect TSS accumulation on a variety-dependent basis, and the authors observed a significant but marginal increase in pH in grapes from vines exposed to CT. The increase in pH was similar in absolute terms to the average decrease in TA [30,32]. However, the decrease in TA was not significant, although several individual studies reported occasional significant decreases in TA in vines exposed to CT [33,34]. The meta-analysis and individual studies also showed that the concentration of anthocyanins and phenols in wine did not increase significantly because of CT [30,31]. However, many individual studies have indicated a positive effect of CT on the anthocyanin and phenol concentration of the berries [31,32,33,35]. The influence of CT on the composition of volatile organic compounds (VOCs) in wine and the sensory characteristics has been studied less than other parameters of wine composition. The positive effect of cluster thinning (approximately 40% of the vine’s clusters and 27% in the yield per vine) on the VOCs in the wines and the clear separation of control and thinned wines was also demonstrated with the Syrah grape variety in a Mediterranean climate [34]. In this study, different esters were found in higher amounts in thinned wines. Our earlier study on Welschriesling in a cool climate region also showed that the concentration of the varietal thiols tends to be higher in thinned wines [36]. In addition, studies on cluster thinning in the Syrah and Pinot Noir grape varieties showed that the wines obtained from the thinned vines differ sufficiently from the control wines to be significantly distinguished by a trained panel in a triangle test [33,37].

The aim of this two-year study was therefore to investigate in a simultaneous experimental setup which of the three factors (vineyard location, crop load per vine, and type of alcoholic fermentation) plays a more significant role in the composition of young Merlot wines in terms of the VOC composition and sensory characteristics.

## 2. Materials and Methods

### 2.1. Experimental Design

A two-year trial was carried out in 2020 and 2021 at two locations (B and M), which served to compare two terroirs.

At both locations and in both years, a cluster thinning intervention was carried out with the aim of reducing the yield by about 40%. The fruit of both treatments (control—C; cluster thinning—CT) was vinified by inoculated alcoholic fermentation (iAF).

A portion of the grapes from the above control treatment (C) was vinified by spontaneous alcoholic fermentation (sAF), comparing the results with the above iAF of the C. This part of the experiment aimed to investigate the influence of the type of alcoholic fermentation on the composition of the wine. Both types of alcoholic fermentation were carried out in both years for location M, but only in 2021 for location B.

### 2.2. Vineyards

#### 2.2.1. Locations and Grapevine Variety

Location B is in Bukovje, Bizeljsko, Slovenia (46°1′38.23″ N, 15°41′38.90″ E), in the Bizeljsko-Sremič wine-growing region. The vines were planted in 2012 at a spacing of 2.4 m (row) × 0.9 m (vine) in an NS row orientation in a silty–loamy soil. Location M is in Mačkovci, Prekmurje, Slovenia (46°46′58.23″ N, 16°9′13.33″ E), in the Štajerska Slovenija wine-growing region. The vines were planted in 2007 in sandy–loamy soils at a spacing of 2.4 m (row) × 0.9 m (vine) in an NS row orientation. The vines were trained at site M on a single Guyot with an average of seven buds per cane and at location B on a cordon with an average of eight buds on four spurs. The height and width of the canopy were the same at both locations, namely 1.2 × 0.3 m.

The vineyards at both locations are planted with *Vitis vinifera* L. cv. Merlot grafted on SO4 rootstock (*Vitis berlandieri* × *Vitis riparia*) and were not irrigated. During the growth periods, the phenology of the vines was monitored using the BBCH scale [38]. The nutrition, pest control, and other vineyard operations were consistent with accepted commercial vineyard practices.

#### 2.2.2. Climatic Parameters

Meteorological data were collected from the Agrometeorological Portal of Slovenia (UVHVVR, Agrometeorološki portal Slovenije, https://agromet.mkgp.gov.si/APP2/Home/Index, accessed on 10 January 2022) from the official weather stations (WSs) near the experimental vineyards (WS Bizeljsko for location M, and WS Selo for location B) to calculate the average temperatures and precipitation for the period from 1 April to 30 September for both years. The growing degree days (GDDs) index was calculated according to the literature reports [39] using 10 °C as the base temperature for the vine subtracted from the average daily temperature recorded from 1 April to 30 September in Appendix A, or from 1 April until the phenological stage of harvest in Appendix A. Regarding the topography of the site, it can be concluded that these data represent the mesoclimates of the two locations.

#### 2.2.3. Reducing the Crop Load and Viticultural Parameters

In both vineyards, the treatments were imposed in a randomised complete block design with 15 replications. After flowering, the number of all the shoots and grapes per vine was counted on about 60 vines per row. In each vineyard, three neighbouring rows (blocks) were included in the experiment. The length of the rows was about 100 m at location B and about 200 m at location M. For the experiment, 30 vines with the most uniform parameters possible were selected in each row, 15 vines per treatment. The exception was location M in 2021, where the number of grapes per vine differed significantly between the C and CT groups even before CT (Appendix A). The vines for the two treatments were distributed randomly and alternately in each row. A total of 45 vines per treatment were included in the trial. There were at least three buffer rows on each side of the experimental plot and five buffer vines at both ends of each row. Crop reduction in the CT treatment was performed at veraison (BBCH 83–85) by CT. All the third, and most second, clusters on each bearing shoot were removed, including the single cluster on the bearing shoot if necessary, representing 27% (location B) and 35% (location M) of the original number of clusters in 2020, and 40% (location B) and 54% (location M) in 2021.

Finally, the number of shoots per metre, the number of clusters per vine at harvest, the yield harvested per vine, and the average weight of the grapes were recorded, calculated, and statistically evaluated in each year. The statistical analyses for location B were performed on 19 vines for each treatment in 2020 and on 17 vines for each treatment in 2021. The statistical analyses for location M were performed on 18 vines for treatment C and 20 vines for treatment CT in 2020 and on 18 vines for each treatment in 2021.

### 2.3. Vinification of Grapes

#### 2.3.1. Inoculated Alcoholic Fermentation (iAF)

The healthy grapes were harvested on 30 September (2020) and 5 October (2021) at location B, and on October 6 in both years at location M. Processing was carried out according to red wine production technology. The grapes (approx. 50 kg) were destemmed using a horizontal stainless steel destemmer and 25 mg/L SO_2_ was added in the form of a 5–6% aqueous solution of sulphurous anhydride. The mash was divided into three 15-L glass fermenters for each treatment. Then, 50 mL of the juice was taken for analysis of the total soluble solids (TSSs), titratable acidity (TAc), and pH. The 0.3 g/L yeast starter culture RX60 (Laffort, Floirac, France) was added directly to each fermenter without prior rehydration of the yeast. Maceration was carried out for 14 days in a temperature-controlled room at 20–22 °C and the cap was soaked twice daily. After an initial sugar drop of about 30%, the yeast nutrients Nutristart and Nutristart Org (Laffort, France) were added to all the AF at a concentration of 0.3 and 0.4 g/L, respectively. The progress of the AF was monitored by refractometric indirect measurements of the density in Oechsle degrees (°Oe). Toward the end of the AF, the residual sugar (g/L) was determined, and the AF was considered complete when the concentration was below 2.0 g/L (Appendix A). After the maceration period, the mash was manually pressed, and the wine was collected in 10-L glass containers. The malolactic bacteria Lactoenos^®^ B7 Direct (Laffort, France) in a quantity of 0.01 g/L and the nutrient Malostart (Laffort, France) in a quantity of 0.3 g/L were added to the wine immediately after pressing. Malolactic fermentation (MLF) was carried out in a temperature-controlled room at 20–22 °C and the malic acid and lactic acid in the wines were then determined. When the malic acid was below 0.1 g/L (between late November and early January), the wines were sulphurised to 50 mg/L SO_2_ (in the form of a 5–6% aqueous solution of sulphurous anhydride), transferred to a room at 12–14 °C to facilitate sedimentation, and bottled in another demijohn after 10–14 days. The standard parameters of the wine and the concentrations of the volatile compounds were analysed in January and February 2021 (for the 2020 vintage) and in January and February 2022 (for the 2021 vintage) in one replicate per each of three biological replications. The wine was bottled in April of the following year after the harvest, 10 bottles of wine at 0.75 L each for each treatment. The sensory analysis of the wine was carried out in the second half of May 2021, approximately one month after bottling of vintage 2020, and in August 2022, approximately three months after bottling of vintage 2021. One bottle per treatment was used for the sensory analysis.

#### 2.3.2. Spontaneous Alcoholic Fermentation (sAF)

A portion of the mash obtained from the grapes of the control crop load (location M) was processed separately before the addition of SO_2_. It was divided into three sterile 3-L glass fermenters and fermented with spontaneous alcoholic fermentation by the indigenous microflora. All the processing equipment that encountered the mash was cleaned with hot high-pressure water to minimise the possibility of cross-contamination. All the other processing steps and conditions, except for the addition of the yeast inoculum, were as described above for the iAF. When the malic acid levels were below 0.1 g/L (between late November and early January), the wines were sulphurised with 50 mg/L SO_2_ (administered as a 5–6% aqueous solution of sulphurous anhydride), transferred to a 12–14 °C room to facilitate sedimentation, and bottled in three 0.75 L glass bottles after one week. The wine analyses were carried out at the same time as described above.

The same experiment was performed at location B, but only for the 2021 vintage. All the AF and MLF were successfully completed in the experiment, as the concentrations of the reducing sugars were below 2.0 g/L and the concentrations of the malic acid were below 0.1 g/L (Appendix A).

### 2.4. Analytical Methods

#### 2.4.1. Analysis of Yeast Population

During the iAF and sAF, the must aliquots were aseptically sampled to determine the composition of the yeast species and to count the colony forming units (CFUs). The samples were diluted and cultured on Wallerstein Laboratory (WL) nutrient agar (Merck KGA, Darmstadt, Germany). The must was sampled and microbiologically analysed after days 0, 3, 6, 11, and 14 of inoculation (iAF) and after days 0, 3, 6, 11, 14, and 18 of sAF. The yeast colonies were classified into the non-*Saccharomyces* and *Saccharomyces* groups according to the macro- and microscopic characteristics described in the literature [40]. The bacterial population was not monitored during the inoculated MLF.

#### 2.4.2. Analysis of Must and Wine Parameters

The total soluble solids (TSSs) in the must were determined with a digital refractometer (WM-7, Altago, Saitama, Japan). The pH of both the must and the wine was determined with a MeterLab PHM 210 (Radiometer Analytical, Lyon, France), and the titratable acidity (TAc) was determined by sodium hydroxide titration and with bromothymol blue as an indicator of colorimetric change (European Commission Regulation (EEC) No. 2676/90, 1990). The yeast assimilable nitrogen in the must was not determined. The ethanol content of the wine was determined with an alcohol meter (Alcolyzer Wine M, Anton Paar, Graz, Austria), and the density of the wine was determined with a density meter (DMA 4500M, Anton Paar, Austria), while the reducing sugars and VA were quantified with an enzymatic robot (BS-200, Mindray, Shenzhen, Guangdong, China). The total dry matter (TDM) of the wine was determined according to the official OIV method [41].

#### 2.4.3. Analysis of Organic Acids

The malic and lactic acids were determined in the wines before and during MLF using a 1100 series liquid chromatograph (HPLC) (Agilent technologies, Palo Alto, CA, USA) connected to a diode array detector (DAD) (Agilent technologies, Palo Alto, CA, USA). The wine samples were filtered through 0.45 µm filters (Merck, Darmstadt, Germany) and injected directly. The organic acids were separated on a Rezek RCM monosaccharide Ca^+2^ cation exchange column (Phenomenex, Torrance, CA, USA). An isocratic technique was used with sulphuric acid (0.0125 mmol/L) as the mobile phase at a constant flow rate of 0.5 mL/min. The injection volume was 10 μL, the column temperature was 65 °C and the time of analysis was 30 min [42].

#### 2.4.4. Analyses of Polyphenols and Anthocyanins

Spectrophotometric analyses were performed using an Agilent 8453 spectrophotometer (Agilent Technologies Inc., Palo Alto, CA, USA) as described previously [43,44]. Prior to analysis, the polar compounds (sugars, free SO_2_, amino acids, and organic acids) were removed from the wines using Sep-Pak C-18 columns (0.5 g, Waters, Milford, MA, USA). The total anthocyanins (TAns) were determined using the absorption maximum in the visible range between 536 and 542 nm and evaluated in mg/L. The total polyphenols (TP), expressed as (+)-catechin in mg/L, were estimated by reduction with the Folin–Ciocalteu reagent to blue pigments due to the phenols in the alkaline solution.

#### 2.4.5. Analysis of Methoxypyrazines

Determination of the methoxypyrazines (IBMP and IPMP) in bottled wines was performed using the method previously described [45]. Wine samples were prepared in 25 mL flasks with the addition of 125 μL of [2H3]-IBMP (internal standard) at 5 μg/L. In a 20 mL SPME vial, 1.6 mL of prepared wine sample, 6.4 mL of Milli-Q water and 2 mL of 4 M NaOH were pipetted, then 3 g of NaCl and a stir bar were added. The closed vials were placed on a magnetic stirrer to dissolve the NaCl. The sample vial was heated for 40 min at 40 °C and the methoxypyrazines were absorbed on divinylbenzene/carboxene/polydimethylsiloxane (DVB/CAR/PDMS) fibre (Supelco, Bellefonte, PA, USA). The methoxypyrazines were identified and quantified using a gas chromatograph (Agilent Technologies 7890A, Santa Clara, CA, USA) equipped with a MPS 2 automatic sampler (Gerstel, Mülheim an der Ruhr, Germany) coupled to a mass spectrometric detector (Agilent Technologies 5975C upgraded with a Triple Axis detector, CA, USA). The chromatographic conditions were as previously described [45]. Calibration was performed using calibration standards in alcoholic solution using IBMP, [2H3]-IBMP and IPMP (Sigma Aldrich, St. Louis, MO, USA). Nine-point calibration (four repetitions per calibration level) was performed for the deployed method using spiked alcoholic solutions. The limit of quantification (LOQ) was 1.2 ng/L for IBMP and 1.6 ng/L for IPMP [45].

#### 2.4.6. Analysis of Varietal Thiols

The thiols in the wines were extracted with Dowex resin according to previously published methods [46], with some slight modifications in terms of sample preparation, as described [45]. In brief, 50 mL of wine sample was spiked with internal standards, i.e., 4-methoxy-2-methyl-2-mercaptobutane (Sigma Aldrich, Schnelldorf, Germany) for 4-mercapto-4-methylpentan-2-one (4MMP) quantification, [2H2]-3-mercaptohexyl acetate (University of Auckland, Auckland, New Zealand) for 3MHA quantification, and [2H2]-3-mercaptohexan-1-ol (University of Auckland, New Zealand) for 3MH quantification. The pH of the wine was adjusted to 7 and the wine was passed through a Dowex column before being eluted with cysteine buffer as described previously [45,46]. The thiols were extracted from the cysteine buffer by liquid–liquid extraction with ethyl acetate and dichloromethane. The organic phase was collected, dried over anhydrous sodium sulphate, and concentrated to a final volume of 50 µL. The samples were analysed using a gas chromatograph (GC) (Agilent Technologies 7890A, CA, USA) coupled to a mass spectrometer (Agilent Technologies 5975C, CA, USA). The chromatograph was equipped with a capillary column (Agilent J&W GC column: HP-INNOWAX, 60 m × 0.25 mm; film thickness 0.25 μm; CA, USA). Helium gas at a constant flow of 0.6 mL/min was used as a carrier. The injector temperature was set to 240 °C with an oven temperature gradient of 50 °C for 5 min, then from 50 °C to 115 °C at 3 °C/min, then from 115 °C to 150 °C at 40 °C/min, 3 min at 150 °C, then from 150 °C to 205 °C at 3 °C/min, from 205 °C to 250 °C at 10 °C/min, 19.625 min at 250 °C, then back from 250 °C to 50 °C at 40 °C/min and 3 min at 50 °C. The ion source temperature was 230 °C, the auxiliary temperature was 250 °C and the quadrupole temperature was 150 °C [45]. One-point calibration was performed using calibration standards in an alcoholic solution with a final concentration of 65 ng/L 4MMP, 650 ng/L 3MHA and 1202 ng/L 3MH, and injected after every ninth sample. The limit of quantification (LOQ) was 2, 4, and 60 ng/L for 4MMP, 3MHA, and 3MH, respectively.

#### 2.4.7. Analysis of Higher Alcohols, Ethyl Acetate, Acetaldehyde, and Diethylacetal

The higher alcohols, ethyl acetate, acetaldehyde, and diethyl acetal were analysed by GC coupled to a flame ionisation detector (FID) (Hewlett Packard 6890, Waldbron, Germany) without prior extraction, using CP-Wax, 57CB, 50 m × 0.25 mm, film thickness 0.20 µm [47]. The wine sample (5 mL) was spiked with the internal standard 4-methyl-2-pentanol (50 μL) (Sigma Aldrich, Schnelldorf, Germany; 2.78 g dissolved in 100 mL of absolute ethanol), shaken, and then 1 µL was injected directly into the GC-FID. Validation of the method was performed [48] and quantification was performed as previously reported [47,48,49].

#### 2.4.8. Analysis of Esters, 1-Hexanol, (Z)-3-Hexenol, γ-Butyrolactone, and Benzyl Alcohol

The esters, C6 alcohols, and other compounds were analysed as previously described [47] after liquid–liquid extraction with dichloromethane and concentration of the organic phase to 1 mL. The wine (100 mL) was transferred into a 250 mL Erlenmeyer flask and cooled down to 0 °C in an ice bath under nitrogen. To this, 23 μg of 4-nonanol was added as an internal standard from the corresponding ethanol solution, using a 50 μL Hamilton syringe. The samples were analysed using a GC (Hewlett Packard 6890, Germany) coupled to an MS (Hewlett Packard 5973, Palo Alto, CA, USA). A CP-Wax 57CB 50 m × 0.25 mm, film thickness 0.20 µm, Varian (Lake Forest, CA, USA) column coupled to a deactivated 2 m × 0.25 mm fused silica guard column (Agilent Technologies, CA, USA) was used. One-point calibration (one concentration level) was performed with a mixture of standards of all the analysed compounds in dichloromethane. For identification, the retention times and mass spectra (scanning was performed in selective ion monitoring mode—SIM mode) were used. For quantification, the peak area of the analysed compound in the sample was multiplied by the concentration of the same compound in the calibration solution and divided by the peak area of the same compound in the calibration solution. The calibration point was scanned every six samples. The results of the minor volatile compounds were corrected according to the concentration factor and the recovery of 4-nonanol [47,48,49].

#### 2.4.9. Sensory Analysis

After the wines had matured for six to nine months at 12 °C, they were evaluated by a group of eight tasters. There were four men and four women, with an age range of 30 to 50 years. The tasters were selected from trained wine tasters with state certification in official wine tasting and are highly experienced in the sensory analysis of Merlot wines. The sensory analysis was conducted in an environment free of noise, visual stimuli and ambient odour in a sensory room with separated booths. Three independent sessions were conducted for each sensory test. The wines (30 mL) were served simultaneously at 15 °C in International Standard Organization wine-tasting glasses. In the first comparison, wines from the control versus reduced yield (C vs. CT) were compared independently for both vineyard locations. In the second comparison, wines produced with inoculated alcoholic fermentation were compared with wines produced with spontaneous alcoholic fermentation independently for both vineyard locations. In the third comparison, the wines from both vineyard locations (M vs. B) were compared independently of each other for both yields (C and CT).

A triangle test was performed to confirm or reject the sensory differences between the two samples. The order of the wines in each session was randomised for each assessor and each test according to a randomised three-digit number for identification. The critical number of correct responses in a triangle test at the indicated α–level for the corresponding number of responses (*n* = 8) was taken from the table [37,50].

#### 2.4.10. Statistical Analysis

The Shapiro–Wilk normality test was used to check the normality of the data, while the homogeneity of variances was tested using the Leven test. Significance was tested using parametric analysis of variance (ANOVA) for variables with normal distribution, and means were separated using Stats–Fisher’s LSD test. For the identified variables with non-homogeneous variances and non-normal distribution, the non-parametric Kruskal–Wallis test was used, and the means were separated using the Bonferroni test. Some missing data for M-C sAF in 2020 (TDM, TAc, pH, TP, Tan, and 1-butanol) were replaced by the mean values for each parameter. When only two groups of samples were compared, significance was tested using the parametric Student’s *t*-test. For the identified variables with non-homogeneous variances and a non-normal distribution, the non-parametric Mann–Whitney test was used. Different letters indicate significant differences at *p* ≤ 0.05. Asterisks indicate the significance level: * *p* ≤ 0.05, ** *p* ≤ 0.01 and *** *p* ≤ 0.001. The unit variation and mean centring were used to create the PCA. The statistical analyses were performed with XLSTAT software version 2023.3.0 [51].

## 3. Results

### 3.1. Climatic Parameters, Development of the Phenological Stages and Grape Ripening

The monthly average temperatures, precipitation, and GDDs index for both vintages (2020, 2021) and for the two vineyard locations (B and M) are shown in Appendix A. The monthly average temperatures and total precipitation for the period from 1 April to 30 September differ between the two years for both vineyard locations (17.4 ± 4.3 °C vs. 16.9 ± 6.1 °C, 574.4 mm vs. 538.2 mm for location B, and 16.8 ± 4.0 °C vs. 16.1 ± 5.7 °C, 583.9 vs. 528.2 for location M). The 2020 vintage was characterised by higher monthly average temperatures in April, May, and August at both vineyard locations, which was reflected in the higher GDDs values for these months. In contrast, the 2021 vintage was characterised by higher monthly average temperatures in June and July, with higher GDDs values (Appendix A, right graphs). In September, a comparable GDDs summation was observed for both vintages at both vineyard locations. The 2020 vintage was wetter from June to September at both vineyard locations, while the 2021 vintage was wetter in April and May (Appendix A, left graphs). The average monthly temperatures from April to June were comparable between the locations in both years. Location B had higher monthly average temperatures and GDDs values from June to September (Appendix A), which was also reflected in a higher GDDs sum for the period from 1 April to 30 September (1658 (location B) vs. 1521 (location M) in 2020; 1647 (location B) vs. 1469 (location M) in 2021). Location B received more precipitation in the spring months, with a peak in May in both years, while location M received more precipitation in June and August in both years and in September 2021 (Appendix A).

The differences in the climatic parameters resulted in slight differences in the occurrence of phenological stages between the years. For example, at location B, the phenological stage at which the fruit berries develop (BBCH 71–79) occurred earlier in 2021 compared to 2020 (Appendix A), and exactly the opposite at location M (Appendix A). The differences in the sum of GDDs between the two locations and between the two years within the same location were even more pronounced. The sum of GDDs at harvest was much higher at location B compared to location M, namely 86 units higher in 2020 and 172 units higher in 2021. In 2021, the GDDs value at harvest was 54 units higher at location B compared to 2020, and 32 units lower at location M (Appendix A). The criteria for harvesting at both locations were that the pH value of the grape should not rise above 3.5 and that the TSS content in the grape should reach at least 20 °Bx.

### 3.2. Grape Yield per Vine and Must Parameters

Appendix A shows that the vines selected for the trial were uniform in terms of the viticultural parameters, namely the number of shoots per metre and the number of clusters per vine before thinning. The only exception was the number of clusters per vine before thinning at location M in 2021. Nevertheless, even in this case, the number of grapes per vine was reduced by 42% compared to the original number. CT had no significant effect on the weight of the bunches at either crop load but, at both locations and in both vintages, it caused a significant decrease in the yield per vine. In 2020, thinning reduced the yield per vine by 27% (location B) and 35% (location M) compared to the control. In 2021, the reduction in the yield per vine due to thinning was even greater, namely 40% (location B) and 54% (location M). Location B recorded a much lower yield in 2021 than in the previous year (Appendix A). The reason for this was the spring frost, which reduced the bunch weight. However, it can be seen from the results that this did not cause a greater vintage effect at this location than at location M (Table 1 and Table 2).

The fresh mass of 100 berries and the must parameters for both locations and years are listed in Appendix A. CT had no significant effect on the fresh mass of 100 berries. CT also had no significant effect on the TSS content (°Bx), TAc content (g/L), or pH of the must, although there was a clear trend toward a higher TSS content and pH at a lower crop load (CT). The results also showed a tendency toward a higher TSS content in the must from location B compared to location M. The influence of the vintage on the quantified parameters was greater and statistically significant at location M.

### 3.3. Fermentation Kinetic and Yeast Population Dynamic in Inoculated and Spontaneous Alcoholic Fermentations (AFs)

#### 3.3.1. Fermentations at Location M

Both types of AF were compared in two consecutive vintages (2020, 2021) for location M. The changes in the yeast cell concentrations and yeast species diversity during spontaneous (sAF) and inoculated (iAF) AF of the Merlot mash are shown in Appendix A. The initial yeast concentration varied between 4.8 and 7.2 log (CFU/mL) depending on the vintage. The duration of both AFs in the 2020 vintage was shorter (14 days for iAF and 18 days for sAF) than in the 2021 vintage (21 days for both AF), most likely due to a higher initial yeast concentration in the 2020 vintage mash (6.7 (sAF) and 7.2 (iAF) log (CFU/mL) in 2020 vs. 4.8 (iAF) and 5.8 (sAF) log (CFU/mL) in 2021). Similar kinetics of AF were observed for the sAF and iAF for each vintage, although the initial yeast population in the iAF was significantly higher in 2020 and significantly lower in 2021 (*p* ≤ 0.05) (Appendix A, upper plots). The dynamics of the yeast populations during the two AFs were assessed by colony morphology and microscopic examination and should serve to group the yeasts into genera. Identification as a species would only be a guess. The highest species diversity was observed at the beginning of the two AFs in the two consecutive years (day 0, Appendix A), with *Hanseniaspora* sp. dominating the yeast population. The abundance of *Hanseniaspora* sp. on day 0 was higher in the sAF compared to the iAF in both vintages (71.5% vs. 53.6% in 2020, 91.2% vs. 76% in 2021), which can be attributed to the inoculation of the commercial strain *S. cerevisiae* (*Saccharomyces* spp.) in the mash of the iAF (Appendix A, middle and bottom graphs). However, yeasts of other genera were present in lower percentages, with the species diversity and abundance differing between vintages. Yeasts of the genus *Saccharomyces* spp. (8.1% (2020) to 9.8% (2021) in sAF, 19.7% in the iAF of both vintages) and *Pichia* spp. (4.0% (2020) and 0.6% (2021) in sAF; 3.3% (2020) and not detected (2021) in iAF) were also isolated from the must of both vintages on day 0. Yeasts of the genus *Starmerella* were only detected in the 2020 vintage (32.5% in sAF, 6.2% in iAF) and of the genus *Metschnikowia* in the 2021 vintage (0.17% in sAF) (Appendix A, middle and bottom graphs). As alcoholic fermentation progressed, the species diversity decreased in both AFs. However, in the sAF of both vintages, *Hanseniaspora* sp. yeasts dominated in the first six days, while *Saccharomyces* spp. yeasts developed thereafter and dominated until the end of the sAF. The main difference between the two vintages in the sAF was that the yeast *Hanseniaspora* sp. was also detected at the end of this AF of the 2020 vintage (days 14 and 18). In contrast to the sAF, yeasts of the genus *Saccharomyces* already predominated in the iAF of both vintages on the third day (Appendix A, lower diagrams).

#### 3.3.2. Fermentations at Location B

Both types of AF were also compared for location B, but only in the 2021 vintage (Appendix A). Comparable fermentation kinetics were observed in both AFs, although the cell concentration in the iAF was significantly higher in the period from day 3 to day 10 (*p* ≤ 0.05), which can be attributed to the inoculation of the mash with a commercial *S. cerevisiae* strain (Appendix A, upper graph). During both AFs, the same trend in the yeast population dynamics was observed as in the mash from location M. *Hanseniaspora* sp. dominated the yeast population until day 6 (sAF) or only until day 0 (iAF), with *Saccharomyces* spp. predominating from day 3 of iAF and day 10 of sAF. Yeasts of other genera were present in low abundance at the first sampling of the mash (day 0), namely *Pichia* spp. (0.9% in sAF, 0.8% in iAF), *Starmerella* sp. (0.3% in sAF, 0.9% in iAF) and *Metschnikowia* sp. (2.8% in sAF, 2.0% in iAF). Their abundance decreased in the following three samplings of both AFs (days 3, 6, and 10) (Appendix A, middle and bottom graphs).

### 3.4. Influence of Cluster Thinning and Type of Alcoholic Fermentation on Wine Parameters

#### 3.4.1. Location M

Table 1 shows the results of the influence of cluster thinning (CT) and the type of alcoholic fermentation (AF) on the volatile compounds and other wine parameters for location M. The influence of CT on the Merlot variety showed a significant and repeatable year-on-year effect on the increase in the ethyl acetate, volatile acidity, and pH of the wine. In 2020 and 2021, CT increased the pH value of the wine from 3.61 to 3.64 and from 3.60 to 3.65, the volatile acidity of the wines from 0.35 to 0.40 g/L and from 0.29 to 0.35 g/L, and the concentration of ethyl acetate from 54.7 to 70.4 μg/L and from 33.9 to 46.5 μg/L.

For the IBMP, ethyl dodecanoate, ethyl palmitate, sum of EEFAs, isoamyl acetate, 2-phenethyl acetate, sum of acetates, 1-hexanol, (Z)-3-hexen-1-ol, 1-propanol and TAns parameters, a significant interaction between CT and vintage was confirmed, with CT causing an increase in one year and a decrease in another year in their concentrations. It is also evident that the vintage influenced more significant differences in the levels of the individual compounds than CT itself, as significantly higher levels of ethanol, 3MH, and the sum of acetates and other alcohols than ethanol and 3MH and significantly lower levels of IBMP, the sum of EEFAs, diethyl succinate, γ-butyrolactone, and ethyl acetate were quantified in the 2021 wines (Table 1).

The influence of the AF type on the Merlot variety at location M was demonstrated in two consecutive years, as spontaneous AF had a significant and reproducible effect on the decrease in the ethyl decanoate content and the increase in the sum of acetates and 2-phenylethanol content (Table 1). In 2020 and 2021, spontaneous AF reduced the content of ethyl decanoate from 52.0 to 36.3 μg/L and from 56.3 to 13.2 μg/L and increased the sum of acetates from 30.0 to 95.3 μg/L and from 43.2 to 70.9 μg/L and the content of 2-phenylethanol from 38.9 to 56.8 mg/L and from 37.6 to 57.6 mg/L (Table 1). For the 3MHA, 1-propanol, and TP parameters, a significant interaction between the type of AF and the vintage was confirmed. All the other influences depended on the vintage. We found significantly higher levels of 2-phenethyl acetate and 2-methyl-1-propanol, and significantly lower levels of (Z)-3-hexen-1-ol, in the wines from sAF in 2020. This comparison also showed that the vintage triggered more significant differences in the levels of the individual compounds than the type of AF itself (Table 1).

A PCA was applied to the quantified wine parameters over the two vintages for location M and explained 65.69% of the variation with the first two dimensions (Figure 1). The M-C and M-CT treatments were not separated by the PCA analysis in each experimental year, but principal component (PC) 1 separated both vintages (Figure 1a), with the 2021 vintage M-C and M-CT wines positively associated with ethanol, total acidity, 2-methyl-1-butanol, and benzaldehyde, and the 2020 vintage M-C and M-CT wines positively associated with 4MMP, 3MHA, ethyl decanoate, (Z)-3-hexen-1-ol, and TP (Figure 1b).

Figure 1a also shows that the M-C wines of the sAF (M-C_sAF) differed from the wines of the M-C treatment with inoculated fermentations by PC2 and explained 26.56% of the variation in the dataset. The imposed 95% confidence ellipses showed that significant differences were observed between the M-C_sAF and M-C (with iAF) treatments in both vintages (Figure 1a). The wine VA, ethyl acetate, ethyl dodecanoate, 2-methyl-1-propanol, 2-phenylethanol, 2-phenethyl acetate, and sum of acetates were positively associated with sAF (Figure 1b). It can be assumed that the type of alcoholic fermentation in both vintages at this location had a greater influence on the wine parameters than the yield reduction due to cluster thinning, and that the vintage effect was less pronounced within the type of alcoholic fermentation than within the different crop loads.

#### 3.4.2. Location B

Table 2 shows the results of the influence of cluster thinning (CT) and the type of alcoholic fermentation on the volatiles and other wine parameters for location B. The influence of CT on the Merlot variety at this location and in two consecutive years was shown to have a significant and repeatable year-on-year effect on the increase in the ethanol content and pH values. In 2020 and 2021, CT increased the ethanol content from 13.46 to 14.09% vol. and from 15.38 to 15.81% vol. and the pH value of the wine from 3.48 to 3.56 and from 3.65 to 3.70. A significant interaction between CT and vintage was confirmed for the ethyl palmitate, benzyl alcohol, (Z)-3-hexen-1-ol, sum of alcohols and VA. The other influences depended on the vintage; otherwise, a significantly lower content of (Z)-3-hexen-1-ol was found in the CT wines in 2020 and a significantly higher content of VA, ethyl palmitate, sum of acetates and alcohols, 1-hexanol, 1-propanol and (Z)-3-hexen-1-ol in 2021, together with a trend to significantly lower content of TP, in the CT wines of both vintages (Table 2). It is obvious that here, too, the vintage has triggered more significant differences in the content of the individual compounds than the CT itself, because in 2021, for example, a significantly higher content of ethanol, VA, pH, ethyl butyrate, ethyl dodecanoate, ethyl palmitate and diethyl succinate and significantly lower levels of IBMP, isoamyl acetate, 1-hexanol, benzyl alcohol, (Z)-3-hexen-1-ol and the sum of alcohols compared to the 2020 vintage were found in at least one of the wines from the two crop loads (C or CT) (Table 2).

In 2021, spontaneous AF had a significant impact on the decrease in the content of ethanol (from 15.38 to 14.72% vol.), 3MH (from 1.305 to 614 ng/L), ethyl butyrate (from 178.7 to 72.9 μg/L), ethyl octanoate (from 232.0 to 86.6 μg/L) and 1-propanol (from 16.2 to 12.8 mg/L), and the the increase in the content of 3MHA (from 8.09 to 14.66 ng/L), 2-phenethyl acetate (from 34.3 to 54.3 μg/L), sum of acetates (from 37.7 to 58.1 μg/L), 2-methyl-1-propanol (from 49.9 to 72.4 mg/L), ethyl acetate (from 53.8 to 78.5 μg/L), VA (from 0.56 to 0.94 g/L) and pH (from 3.65 to 3.78) (Table 2).

A PCA was applied to the quantified wine parameters across two vintages for location B and explained 72.90% of the variation with the first two dimensions (Figure 2). The B-C and B-CT treatments were not separated by the PCA analysis in each experimental year, but PC1 and PC2 separated both vintages (Figure 2a), with the B-C and B-CT wines of the 2021 vintage positively associated with ethanol, 3MH, ethyl butyrate, 1-butanol, ethyl palmitate, and ethyl dodecanoate, and the B-C and B-CT wines of the 2020 vintage positively associated with IBMP, 4MMP, 3MHA, ethyl decanoate, hexyl acetate, isoamyl acetate, 1-hexanol, benzyl alcohol, (Z)-3-hexen-1-ol, 3-methyl-1-butanol, sum of alcohols, and TAns (Figure 2b). Figure 2a also shows that the 2021 B-C wines of the sAF (B-C_sAF 2021) differ from the 2021 wines of the B-C treatment with inoculated fermentations (B-C_iAF 2021) by PC2, which explains 20.36% of the variation in the dataset. However, the imposed 95% confidence ellipses showed no significant differences between treatments B-C_sAF and B-C_iAF 2021 (Figure 2a), but it should be emphasised that at this location, sAF was only performed in 2021. The wine’s content of 2-methyl-1-propanol, ethyl acetate, diethyl succinate, benzaldehyde, γ-butyrolactone, VA, TDM and pH were positively associated with sAF (Figure 2b).

#### 3.4.3. Comparison of Locations M and B

The influence of the location on the significant differences in the content of volatiles and some other compounds in the Merlot wine of inoculated alcoholic fermentations in two consecutive years and at two crop loads is shown in Table 3. The differences are more reproducible within the same year at both crop loads than within the same crop load at two consecutive vintages. The wines from both locations in 2020 differed significantly in their content of ethanol, IBMP, 3MH, ethyl decanoate, sum acetates, 2-methyl-1-butanol, 3-methyl-1-butanol, 2-phenylethanol, and sum of alcohols (higher content at location B), and 1-hexanol and ethyl acetate (higher content at location M) at both crop loads (C and CT). In the same comparison in 2021, there were fewer significant differences, namely that the wines from both locations, regardless of the crop load, differed in their content of VA, TP, 3MH, ethyl palmitate, γ-butyrolactone, ethyl acetate (higher content at location B), and hexyl acetate (higher content at location M). When comparing the same crop load in two consecutive vintages, it was found that the wines from both locations differed significantly in the ethanol and 3MH content at a higher crop load (C), regardless of the vintage (higher content in location B). At a lower crop load (CT), the wines from both locations differed significantly in TP, 3MH, ethyl octanoate, sum of acetates and alcohols, and 3-methyl-1-butanol (higher content at location B), and 1-hexanol (higher content at location M), irrespective of the vintage (Table 3). Therefore, when comparing the two locations, greater differences were found in the concentrations of volatiles and some other compounds in the wines with a lower crop load (CT).

The only compound that differed between the two locations regardless of the vintage and crop load was 3MH, with significantly higher concentrations in the wines from location B (in 2020 with 777.8 (C) and 688.7 ng/L (CT) compared to the 2020 wines from the location M with 505.1 (C) and 498.8 ng/L (CT) and in 2021 with 1305.0 (C) and 1293.0 ng/L (CT) compared to the 2021 wines from the location M with 668.7 (C) and 562.7 ng/L (CT)) (Table 1, Table 2 and Table 3). There was also a trend toward a higher sum of ethyl esters (EEFAs) (for approx. 30% at the control load and 60% at cluster thinning) and a significantly higher concentration of γ-butyrolactone (for approx. 80% at the control load and 60% at cluster thinning) in the 2021 wines from location B (Table 1 and Table 2). 

PCA was applied to the quantified wine parameters across the two vintages to separate the two locations and explained 64.58% of the variation with the first two dimensions (Figure 3). The 2020 wines from locations B and M were separated by PC2, explaining 26.81% of the variation in the dataset. The 2020 B-C and B-CT wines were positively associated with IBMP, 3-methyl-1-butanol, sum alcohols, and TAns (Figure 3b). The 2021 wines from locations B and M were separated by PC1, explaining 37.77% of the variation in the dataset. The 2021 B-C and B-CT wines were positively associated with ethanol, TDM, 3MH, ethyl decanoate, γ-butyrolactone, and TP (Figure 3b). The M-C and M-CT wines of both vintages were positively associated with (Z)-3-hexen-1-ol, and in 2021, also with TAc. In addition, the imposed 95% confidence ellipses showed that significant differences were observed between the M and B locations for the quantified wine parameters (Figure 3a).

### 3.5. Comparisons of Wines by Sensory Analysis

The sensory analysis was carried out using the triangle test. The results are shown in Table 4. Using the triangle test, the assessors were unable to distinguish between the wines of the two crop loads at either of the two locations in any of the trial years (Table 4). When comparing the two types of AF, inoculated and spontaneous, the opposite was the case. In all cases (location M both vintages and location B vintage 2021), the sensory panel distinguished between the wines from sAF and iAF (Table 4). It is also obvious that the sensory evaluation also revealed statistically significant differences between the wines from both locations, for both crop loads, in both years (Table 4).

## 4. Discussion

The location of the vineyard, with environmental factors such as the soil and climate, is part of the concept of wine terroir, which relates to the sensory characteristics of the wine [56] and is decisive in determining the overall quality of the wine. It was also confirmed that a typical vineyard is defined by its most common climatic characteristics, such as the seasonal sum of temperatures and the water balance, rather than by the classification of soils according to their ability to supply the vines with water [3]. The differences between the wines from the two locations can therefore also be attributed to the higher sum of GDDs in location B in both years, and the lower precipitation at this location during grape ripening. We can also conclude that the tendency toward a higher total anthocyanin content, the higher content of 3MH, some ethyl esters and acetates and γ-butyrolactone (all belonging to the aroma vectors for fruity aromas [53] and all exceeding their OT values for wine or reach at least 0.2 values of its OT (γ-butyrolactone)) and a higher ethanol content and a tendency toward a higher total dry matter (TDM) content in the wines from location B contributed to the significant differences between the wines from both locations. The concentrations of VOCs are normalised by the corresponding odour thresholds (OTs) to yield the odour activity values (OAVs). Aroma vectors with summed-OAVs (or OAV in the case of an aroma vector composed of an individual compound) inferior to 0.2 are probably non-perceptible from a sensory point of view [53]. The most important aroma vectors reaching this value in the wines in our study were thus thiols, ethyl esters (vector for fruity aroma), higher alcohols (vector for alcoholic, solvent aromas) and acetaldehyde (at low levels, it can play positive roles in some specific aromatic contexts like enhancing fruitiness and freshness [54]). If we further consider the differences in the microbiome of grapes from different terroirs with respect to non-*Saccharomyces* and *Saccharomyces* yeasts [9,10,57], even small differences in the non-*Saccharomyces* microbiome at early stages of inoculated alcoholic fermentation of the mash from the control yield of the 2021 vintage and the possible involvement of different *Saccharomyces* strains, which were not determined in our experiment, may have had some influence on the differences in the sensory characteristics of the wine from the two locations.

Within the same location in the experiment, there were more significant differences due to the type of alcoholic fermentation than the yield level. This made it possible to distinguish the wines from the spontaneous and inoculated alcoholic fermentations in both locations by sensory analysis. Our spontaneous alcoholic fermentations were dominated by non-*Saccharomyces* yeasts in the first six days, as per the results previously shown by others [8,9,13,14,15,16]. The influence of these yeast strains was later confirmed by targeted metabolic profiling of the wines, which revealed a significantly higher concentration of 2-phenethyl acetate (vector for flowery aroma), which relates to the sum of acetates, and a significantly higher concentration of 2-phenylethanol (vector for alcoholic, solvent aromas) in the spontaneously fermented wines in all three (sum of acetates) or at least two comparisons (2-phenethyl acetate and 2-phenylethanol) carried out. The results for 2-phenethyl acetate and 2-phenylethanol are consistent with those of others [58]. The tendency toward higher concentrations of higher alcohols, ethyl acetate and volatile acidity was also observed in the spontaneously fermented wines, and a significant difference was confirmed for location B in 2021. The above-mentioned metabolites are increasingly produced by many *Hanseniaspora* strains and are therefore often found in higher concentrations in the wines from spontaneous alcoholic fermentation [18,20,21] or mixed fermentations of *S. cerevisiae* with *H. uvarum* [59] or with *H. occidentalis* [60]. In our study, the trend of a higher acetaldehyde content in the spontaneous alcoholic fermentations could also influence the formation of stable pigments and pyranoanthocyanin adducts, such as vitisin A and B in red wine, as already reported [28,61]. The larger population of *Starmerella* sp. yeast in sAF (location M) may have had some influence on the production of 3MH [26] and benzyl alcohol [62], with the observed trend toward lower concentrations of 3MH and higher concentrations of benzyl alcohol in our spontaneous alcoholic fermentations. Yeasts of the genus *Starmerella* are also known to increase the content of lactones in wines [26], and a similar trend was also observed for the γ-butyrolactone concentrations in our study. The trend of a lower content of ethyl esters (EEFAs) in our spontaneously fermented wines can also be noticed, which was also shown by other authors for non-inoculated fermentations on an industrial scale [57]. The concentration of esters in wine depends on the bioactivity of yeast’s ester-synthesizing enzymes and esterases [63]. The balance between both processes, together with other enzymes’ activity (lipases), depends on population of *Saccharomyces* and non-*Saccharomyces* strains during the vinification process [64]. Interestingly, we quantified comparable levels of ethyl esters in all three spontaneous fermentations in our study, regardless of the different location and the two vintages. Using a mixed inoculation of *H. uvarum* and *S. cerevisiae*, a higher content of ethyl esters was obtained [19], but our results suggest that regardless of the use of mixed starter cultures of *Saccharomyces* and non-*Saccharomyces* yeasts, spontaneous alcoholic fermentation is not equivalent. The importance of spontaneous wine fermentations in producing VOCs has already been demonstrated in the past [9,16,57]. However, there are few studies that would confirm or refute the influence of this process on the sensory quality of wine. Our study complements this part by showing that the spontaneous alcoholic fermentation has an influence on the sensory characteristic of wine. However, further research is needed to investigate how this can be integrated into the understanding of wine typicity regarding geographical indications or terroir, as was demonstrated for Douro wines [57] and discussed by other authors [8,11].

On the other hand, it was not possible to distinguish the wines with cluster thinning from the wines without crop regulation by means of a sensory triangle test, a result that contradicts the result of the one-year and one-location study on the Syrah variety [33] and the three-year and one-location study on the Pinot Noir variety [37]. Thus, it cannot be confirmed that yield regulation between 27 and 54% of the original yields in two vineyards and two years had a significant influence on the sensory characteristics of the Merlot wine in our study. It is true that in our experiment we could not ensure fully repeatable yields per vine in two consecutive years, as in 2021 the yield at location B was lower due to spring frost, and in the same year at location M, there were differences in the number of clusters between the control and thinned vines even before thinning. However, based on the results of other experiments on the influence of cluster thinning on the sensory quality of the wine, we can conclude that thinning could have a linear effect over a larger range of yield per vine [65]. The chemical parameters showed a tendency toward an increase in the content of ethyl acetate and volatile acids, as well as γ-butyrolactone, in the cluster thinned wines. These differences were smaller than those observed by other authors for the Syrah grape variety, where the esters were found in higher concentrations in the cluster thinned wines [34]. The cluster thinning in the Welschriesling grape variety showed an effect on the significantly increased content of 3MHA in one of the three years of the trial and in some cases also a tendency to increase the content of 3MH [36], which we could not confirm in the current study. At the same time, we also confirmed the results of the meta-study [31] and specific studies [30,34,37] that cluster thinning influences increasing the pH value, with a simultaneous tendency toward a slight decrease in the content of total acids in the wine. These results and the result confirming that CT has no influence on the total anthocyanin content of wine are also consistent with the results of another study on the Merlot grape variety [30] and are in contrast to those obtained for the Pinot Noir grape variety [37]. Our results are also in line with the conclusions of the authors of the recent review on cluster thinning [66] and other studies, in which the authors concluded that the effect of this technical factor on wine composition is an interaction between the variety [67], environment, vintage and intensity of cluster thinning [37], and therefore, the outcome of the intervention is difficult to predict.

It should also be emphasised that in our experiment, the vintage effect outweighed the effect of cluster thinning and the type of alcoholic fermentation. The interaction between the type of alcoholic fermentation and the vintage and also the cultivar was already shown by others [68]. However, spontaneous alcoholic fermentation reduced the influence of vintage on the wine composition in our study, while the opposite was true for cluster thinning when we considered the first two PCA plots, one for each location. The importance of vintage has already been emphasised, with its effects on the metabolic profiles of grapes outweighing the effects of soil [3] as an important factor in determining wine quality.

## 5. Conclusions

So far, at least to our best knowledge, there has been no research on Merlot or any other grape variety that has simultaneously investigated the influence of the environmental and technical factors of cluster thinning and the type of alcoholic fermentation on the composition of volatile compounds (VOCs) and on the sensory characteristics of the wine in successive vintages. Testing all three factors on grapes of the same origin and two vintages has significant added value compared to separate trials for each factor and provides important conclusions for the industry. All the above factors can have a significant influence on the wine composition, as has been shown separately in many other studies and already mentioned, but the results obtained in our study are an important contribution to classifying their influence and understanding their interaction with the vintage effect. Furthermore, the influence of cluster thinning on the composition of VOCs and the sensory characteristics of the wine in more than one-year experiments has been studied far less than its influence on other parameters of wine and fruit composition, so our study also adds some original insights to this area of research, clearly showing that under our experimental conditions it had only a minor influence on the quantified VOCs in wine. The following main conclusions can be drawn from the results: (i) of the three factors examined in our experiment, the composition of Merlot wine was largely determined by the vineyard location.; (ii) spontaneous alcoholic fermentation has a greater influence on the wine composition than cluster thinning; (iii) both the location of the vineyard and spontaneous alcoholic fermentation influenced significant differences in the sensory characteristics of the wine, while cluster thinning had no such influence; and (iv) spontaneous alcoholic fermentations can reduce the effects of vintage on the wine composition, while the opposite is true for cluster thinning.

## Figures and Tables

**Figure 1 foods-14-01101-f001:**
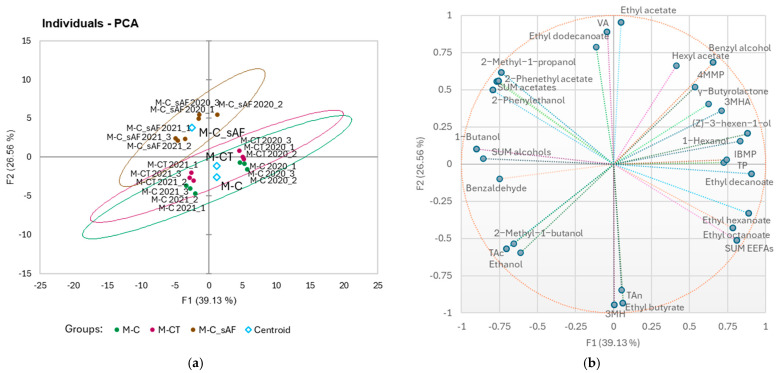
Principal component analyses (PCAs) conducted on the quantified wine parameters for the samples from location M, two vintages (2020, 2021), and for the first two principal components: (**a**) score plot for the first two principal components; ellipses represent 95% confidence intervals for sample groups; M-C—control crop load, M-CT—cluster thinning, both with inoculated alcoholic fermentation; M-C_sAF—control crop load and spontaneous alcoholic fermentation (AF); and (**b**) loadings for the first two principal components; size of the bubbles indicates the importance to the separation of samples for individual variable.

**Figure 2 foods-14-01101-f002:**
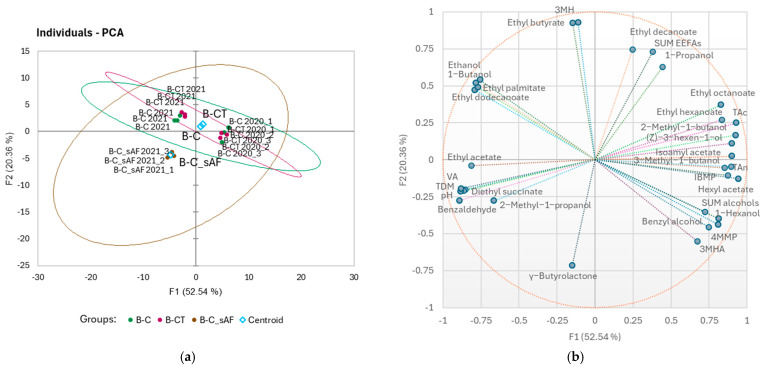
Principal component analyses (PCAs) conducted on the quantified wine parameters for the samples from location B, two vintages (2020, 2021), and for the first two principal components: (**a**) score plot for the first two principal components; ellipses represent 95% confidence intervals for sample groups; B-C—control crop load, B-CT—cluster thinning, both with inoculated alcoholic fermentation; B-C_sAF—control crop load and spontaneous alcoholic fermentation AF (only vintage 2021); and (**b**) loadings for the first two principal components; size of the bubbles indicates the importance to the separation of samples for individual variable.

**Figure 3 foods-14-01101-f003:**
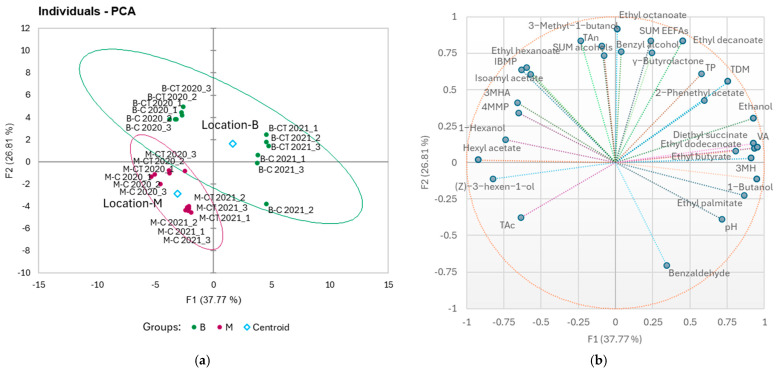
Principal component analyses (PCAs) conducted on the quantified wine parameters for the samples from two locations (M, B) at two crop loads (C, CT), and two vintages (2020, 2021) for the first two principal components: (**a**) score plot for the first two principal components; ellipses represent 95% confidence intervals for sample groups; B—location B, M—location M; and (**b**) loadings for the first two principal components; size of the bubbles indicates the importance to the separation of samples for individual variable.

**Table 1 foods-14-01101-t001:** Effects of the crop load (C—control, CT—cluster thinning; left side) and type of AF (iAF—inoculated alcoholic fermentation, sAF—spontaneous alcoholic fermentation; right side) on the Merlot wine volatiles and other compounds at location M, two vintages (2020, 2021).

	Vintage/Crop Load (C, CT) ^1^				*p* Values ^2^			Vintage/Fermentation Type		*p* Values		
	2020		2021					2020	2021			
	M-C (M-C_iAF)	M-CT	M-C (M-C_iAF)	M-CT	T	V	T × V	M-C_sAF ^3^	M-C_sAF	T	V	T × V
Methoxypyrazine (ng/L)												
IBMP (6–15 ^4^)	2.10 ± 0.10	2.47 ± 0.15	1.93 ± 0.07	1.82 ± 0.08	ns	***	**	1.97 ± 15	1.50 ± 0.35	ns	ns	ns
Varietal thiols (ng/L)												
4MMP (0.8)	11.56 ± 5.2	8.69 ± 3.17	3.74 ± 0.81	3.93 ± 1.24	ns	ns	ns	9.6 ± 3.9	6.78 ± 3.38	ns	*	ns
3MHA (0.0042)	24.3 ± 6.1	17.0 ± 4.6	7.13 ± 0.45	7.51 ± 2.01	ns	ns	ns	12.7 ± 3.8	11.27 ± 3.20	ns	**	**
3MH (60)	505.1 ± 39.3	498.8 ± 44.5	668.7 ± 34.3	562.7 ± 63.4	ns	**	ns	313.6 ± 17.7	398.3 ± 6.4	ns	*	ns
Ethyl esters of strain chain fatty acids (EEFAs) (μg/L)												
Ethyl butyrate (20)	120.3 ± 2.5	116.0 ± 6.0	155.2 ± 2.1	125.3 ± 23.4	ns	ns	ns	43.7 ± 7.1	80.7 ± 14.2	ns	*	ns
Ethyl decanoate (200)	52.0 ± 3.6	56.3 ± 4.7	29.0 ± 2.0	23.7 ± 0.2	ns	*	ns	36.3 ± 9.6 b	13.2 ± 2.5 c	***	***	ns
Ethyl octanoate (2)	259.0 ± 13.1	251.0 ± 2.6	181.3 ± 7.3	162.4 ± 0.3	ns	*	ns	125.7 ± 49.2	125.7 ± 49.2	ns	ns	ns
Ethyl dodecanoate (640)	5.67 ± 0.58	7.00 ± 1.00	5.85 ± 0.72	5.29 ± 0.34	ns	ns	*	9.67 ± 2.08	9.35 ± 2.23	ns	ns	ns
Ethyl hexanoate (5)	269.3 ± 7.4	261.0 ± 1.7	143.3 ± 2.1	142.4 ± 5.3	ns	***	ns	141.3 ± 34.1	82.6 ± 4.8	ns	**	ns
Ethyl palmitate (1.5)	12.7 ± 2.5	28.3 ± 3.8	27.3 ± 4.2	19.0 ± 2.1	ns	ns	*	11.3 ± 0.6	17.9 ± 8.7	ns	ns	ns
SUM EEFAs	719.0 ± 21.0	719.7 ± 9.8	541.9 ± 13.6 b	478.1 ± 28.6 c	*	***	*	367.0 ± 99.4	329.5 ± 58.9	ns	*	ns
Acetates (μg/L)												
Hexyl acetate (670)	6.57 ± 2.68	5.53 ± 1.16	3.86 ± 0.15	3.66 ± 0.34	ns	ns	ns	10.33 ± 1.40	4.17 ± 1.96	ns	**	ns
Isoamyl acetate (30)	3.80 ± 0.20 b	4.63 ± 0.21 a	4.62 ± 0.16	4.58 ± 0.34	*	*	*	6.30 ± 1.06	4.55 ± 1.14	ns	ns	ns
2-Phenethyl acetate (250)	19.7 ± 0.6 c	27.7 ± 1.5 b	34.7 ± 0.5	33.4 ± 0.8	***	***	***	78.7 ± 3.5 a	62.2 ± 17.5	*	ns	ns
SUM acetates	30.0 ± 3.3 c	37.8 ± 0.9 b	43.2 ± 0.6	41.6 ± 1.2	*	***	**	95.3 ± 5.8 a	70.9 ± 19.9 b	***	ns	*
Alcohols (mg/L)												
1-Hexanol (8)	2.74 ± 0.02	2.60 ± 0.02	0.89 ± 0.01	1.01 ± 0.06	ns	***	*	2.53 ± 0.10	0.79 ± 0.06	ns	*	ns
Benzyl alcohol (200)	0.64 ± 0.01	0.66 ± 0.06	0.13 ± 0.003	0.15 ± 0.01	ns	*	ns	0.77 ± 0.19	0.25 ± 0.04	ns	*	ns
(Z)-3-hexen-1-ol (0.4)	0.027 ± 0.002	0.023 ± 0.001	0.014 ± 0.0002	0.017 ± 0.001	ns	***	**	0.023 ± 0.001 b	0.014 ± 0.002	*	***	ns
1-Propanol (-)	19.4 ± 1.1	18.8 ± 0.8	16.6 ± 0.1	20.0 ± 2.0	ns	ns	*	17.3 ± 0.7	18.0 ± 0.7	ns	*	**
2-Methyl-1-propanol (40)	45.7 ± 0.5	51.9 ± 1.6	55.9 ± 0.07	59.3 ± 1.5	ns	*	ns	79.2 ± 3.5 a	74.4 ± 5.9	*	ns	ns
1-Butanol (150)	0.80 ± 0.08	0.78 ± 0.02	0.98 ± 0.13	0.93 ± 0.06	ns	ns	ns	nd ^5^	1.10 ± 0.11	ns	*	ns
2-Methyl-1-butanol (30)	62.4 ± 1.2	61.9 ± 1.1	67.2 ± 0.6	68.8 ± 0.6	ns	***	ns	57.6 ± 3.7	69.6 ± 7.5	ns	ns	ns
3-Methyl-1-butanol (30)	201.6 ± 4.2	204.4 ± 4.3	211.9 ± 1.2	214.2 ± 1.9	ns	ns	ns	186.9 ± 31.4	234.4 ± 43.7	ns	ns	ns
2-Phenylethanol (14)	38.9 ± 4.0	37.6 ± 0.6	46.5 ± 0.8	46.8 ± 1.8	ns	***	ns	56.8 ± 5.6 ab	57.6 ± 8.9 a	**	ns	ns
SUM alcohols	372.2 ± 7.7	378.6 ± 8.2	400.0 ± 1.1	411.2 ± 2.0	ns	*	ns	401.2 ± 36.1	456.1 ± 65.5	ns	ns	ns
Other compounds, pH												
Benzaldehyde (μg/L) (2000)	3.07 ± 0.67	2.73 ± 0.31	5.40 ± 0.89	6.32 ± 0.44	ns	ns	ns	4.07 ± 2.04	7.48 ± 0.80	ns	**	ns
Diethyl succinate (μg/L) (200,000)	496.7 ± 20.8	559.6 ± 99.0	392.3 ± 14.2	466.3 ± 29.6	ns	*	ns	723.3 ± 781.2	534.6 ± 160.9	ns	ns	ns
γ-Butyrolactone (μg/L) (35,000)	7083 ± 361.2 b	7737 ± 358.4 a	4220 ± 77.4	4828 ± 393.2	**	***	ns	6467 ± 2261	6711 ± 764.6	ns	ns	ns
Acetaldehyde (mg/L) (0.5)	7.73 ± 2.83	6.26 ± 2.63	10.5 ± 0.63	8.44 ± 0.58	ns	ns	ns	9.12 ± 4.51	22.0 ± 6.6	ns	ns	ns
Ethyl acetate (μg/L) (7500)	54.7 ± 2.3 b	70.4 ± 3.7 a	33.9 ± 7.7 c	46.5 ± 4.7 b	**	***	ns	167.7 ± 39.3	123.3 ± 7.8	ns	*	ns
Ethanol (% vol)	11.93 ± 0.06	11.97 ± 0.13	12.82 ± 0.06	13.00 ± 0.01	ns	*	ns	11.80 ± 0.10	12.80 ± 0.30	ns	***	ns
TDM (g/L)	27.4 ± 0.2 b	28.1 ± 0.2 a	27.2 ± 0.4	27.3 ± 0.0	*	**	ns	nm ^6^	27.6 ± 0.2	ns	ns	ns
TAc (g/L tartaric acid)	5.80 ± 0.00	5.77 ± 0.06	6.17 ± 0.06	6.13 ± 0.06	ns	ns	ns	5.50 ± 0.00	6.03 ± 0.15	ns	ns	ns
VA (g/L acetic acid)	0.35 ± 0.01 b	0.40 ± 0.03 a	0.29 ± 0.02 c	0.35 ± 0.03 b	**	**	ns	0.53 ± 0.13	0.43 ± 0.04	ns	*	ns
pH	3.61 ± 0.01 b	3.64 ± 0.01 a	3.60 ± 0.01 b	3.65 ± 0.01 a	***	ns	ns	3.63 ± 0.00	3.62 ± 0.03	ns	ns	ns
TP (mg/L (+)-Catechin)	1594 ± 8.9	1617 ± 45.8	1333 ± 108.8	1461 ± 59.9	ns	***	ns	1330 ± 186.9	1487 ± 20.4	ns	ns	**
TAns	709.8 ± 21.3	735.5 ± 30.6	808.4 ± 54.8	728.4 ± 11.6	ns	*	*	552.1 ± 108.6	639.7 ± 31.1	ns	**	ns

^1^ Two-way ANOVA was used to compare data. Means followed by a different letter in a row are significant at *p* ≤ 0.05 (Fisher’s LSD). All the stated uncertainty is a standard deviation of three replicates per treatment, except TAc, pH, TP and TAns at M-C_sAF in 2020, where only two replicates were performed. ^2^ The asterisks indicate the level of significance of the two-way ANOVA for T, treatment, V, vintage and T × V, interaction treatment × vintage: * *p* ≤ 0.05, ** *p* ≤ 0.01 and *** *p* ≤ 0.001, whereas ns indicates no significant differences. ^3^ Column M-C_sAF is compared with column M-C (M-C_iAF) for each vintage. ^4^ The olfactory thresholds (OTs) for the compounds from IBMP to ethyl acetate—the reference for each value was listed previously [52] or taken elsewhere [53,54]. The reported OTs for IBMP were 6 ng/L in the model solution (ethanol at 12% volume, 5 g/L tartaric acid, pH 3.5) and 15 ng/L in Merlot wine [55]. ^5^ nd—not detected. ^6^ nm—not measured.

**Table 2 foods-14-01101-t002:** Effects of the crop load (C—control, CT—cluster thinning; left side) and type of AF (iAF—inoculated alcoholic fermentation, sAF—spontaneous alcoholic fermentation; right side) on the Merlot wine volatiles and other compounds at location B, two vintages (2020, 2021).

	Vintage/Crop Load (C, CT) ^1^				*p* Values ^2^			Vintage/Ferm. Type	*p* Values
	2020		2021					2021	
	B-C	B-CT	B-C (B-C_iAF)	B-CT	T	V	T × V	B-C_sAF ^3^	T
Methoxypyrazine (ng/L)									
IBMP (6–15 ^4^)	5.86 ± 0.21	6.67 ± 1.08	0.73 ± 0.06	1.13 ± 0.15	ns	*	ns	0.89 ± 0.15	ns
Varietal thiols (ng/L)									
4MMP (0.8)	7.42 ± 0.97	10.9 ± 3.7	4.18 ± 1.23	3.27 ± 0.68	ns	*	ns	5.07 ± 2.05	ns
3MHA (0.0042)	19.3 ± 2.0	16.6 ± 1.4	8.09 ± 0.50	9.04 ± 2.36	ns	***	ns	14.66 ± 3.35 a	*
3MH (60)	777.8 ± 26.8	688.7 ± 12.7	1305.0 ± 343.1	1293.0 ± 147.8	ns	ns	ns	614 ± 41.8 b	*
Ethyl esters of strain chain fatty acids (EEFAs) (μg/L)									
Ethyl butyrate (20)	132.3 ± 22.1	128.7 ± 28.7	178.7 ± 22.5	184.3 ± 10.2	ns	**	ns	72.9 ± 10.7 b	**
Ethyl decanoate (200)	71.3 ± 1.5	74.0 ± 1.0	68.0 ± 8.7	78.7 ± 2.1	ns	ns	ns	39.9 ± 1.6	ns
Ethyl octanoate (2)	277.7 ± 5.0	283.0 ± 1.0	232.0 ± 39.0	269.7 ± 12.6	ns	*	ns	86.6 ± 11.0 b	**
Ethyl dodecanoate (640)	5.00 ± 0.00	6.00 ± 1.00	17.3 ± 2.5	18.6 ± 1.2	ns	*	ns	14.4 ± 0.9	ns
Ethyl hexanoate (5)	256.7 ± 7.6	254.7 ± 4.2	163.3 ± 22.8	183.3 ± 4.2	ns	ns	ns	89.7 ± 4.7	ns
Ethyl palmitate (1.5)	5.00 ± 3.46	4.00 ± 1.00	38.0 ± 4.0 b	52.3 ± 7.6 a	*	***	*	30.9 ± 3.1	ns
SUM EEFAs	748.0 ± 26.5	750.3 ± 27.3	697.3 ± 91.4	787.0 ± 16.5	ns	ns	ns	334.4 ± 18.8	ns
Acetates (μg/L)									
Hexyl acetate (670)	5.63 ± 2.29	3.63 ± 1.12	nd ^5^	nd	ns	ns	ns	nd	/
Isoamyl acetate (30)	5.70 ± 0.52	5.83 ± 0.50	3.40 ± 0.61	4.30 ± 0.17	ns	***	ns	3.79 ± 0.43	ns
2-Phenethyl acetate (250)	33.7 ± 0.6	37.3 ± 1.2	34.3 ± 4.0	42.3 ± 2.5	ns	*	ns	54.3 ± 2.0 a	**
SUM acetates	45.0 ± 3.2	46.8 ± 0.3	37.7 ± 4.6 b	46.6 ± 2.7 a	*	ns	ns	58.1 ± 2.0 a	**
Alcohols (mg/L)									
1-Hexanol (8)	1.57 ± 0.02	1.64 ± 0.003	0.79 ± 0.02 c	0.87 ± 0.04 b	*	***	ns	0.92 ± 0.08	ns
Benzyl alcohol (200)	0.76 ± 0.06	0.67 ± 0.006	0.56 ± 0.01	0.58 ± 0.03	ns	***	*	0.62 ± 0.03	ns
(Z)-3-hexen-1-ol (0.4)	0.018 ± 0.001 a	0.014 ± 0.001 b	0.010 ± 0.001 c	0.011 ± 0.0006 d	*	***	***	0.010 ± 0.0003	ns
1-Propanol (-)	19.0 ± 0.1	18.2 ± 0.1	16.2 ± 0.9 b	20.4 ± 0.4 a	*	ns	ns	12.8 ± 1.6 b	*
2-Methyl-1-propanol (40)	48.9 ± 0.2	49.0 ± 0.3	49.9 ± 3.9	54.5 ± 0.2	ns	ns	ns	72.4 ± 3.0 a	**
1-Butanol (150)	0.86 ± 0.04	0.88 ± 0.01	1.55 ± 0.18	1.43 ± 0.09	ns	ns	ns	1.19 ± 0.18	ns
2-Methyl-1-butanol (30)	71.0 ± 0.3	71.1 ± 0.3	63.8 ± 3.0	69.6 ± 0.8	ns	ns	ns	61.9 ± 43.5	ns
3-Methyl-1-butanol (30)	248.2 ± 0.3	248.6 ± 1.0	206.9 ± 10.0	223.8 ± 2.2	ns	ns	ns	213.7 ± 15.8	ns
2-Phenylethanol (14)	49.9 ± 4.4	51.9 ± 1.0	46.4 ± 3.9	51.6 ± 2.7	ns	ns	ns	57.4 ± 7.3	ns
SUM alcohols	440.2 ± 3.9	442.0 ± 1.8	386.2 ± 15.4 c	422.8 ± 5.1 b	**	***	**	421.0 ± 19.3	ns
Other compounds, pH									
Benzaldehyde (μg/L) (2000)	2.17 ± 0.12	2.30 ± 0.52	6.47 ± 4.71	3.17 ± 0.47	ns	ns	ns	27.16 ± 14.95	ns
Diethyl succinate (μg/L) (200,000)	516.7 ± 20.8	480.0 ± 10.0	2220 ± 281.6	2667 ± 245.4	ns	*	ns	5460 ± 2232	ns
γ-Butyrolactone (μg/L) (35,000)	7903 ± 590.8	8466 ± 760.4	7703 ± 196.0	7690 ± 105.4	ns	ns	ns	9569 ± 919.6	ns
Acetaldehyde (mg/L) (0.5)	10.9 ± 1.01	5.75 ± 1.93	14.4 ± 9.3	9.03 ± 2.33	ns	ns	ns	18.1 ± 9.4	ns
Ethyl acetate (μg/L) (7500)	42.0 ± 1.7	44.8 ± 2.4	53.8 ± 6.4	73.0 ± 1.8	ns	*	ns	78.5 ± 10.7 a	*
Ethanol (% vol)	13.46 ± 0.10 d	14.09 ± 0.13 c	15.38 ± 0.02 b	15.81 ± 0.12 a	***	***	ns	14.72 ± 0.26 b	*
TDM (g/L)	29.2 ± 0.2	30.0 ± 0.3	30.8 ± 0.6	30.6 ± 0.8	ns	*	ns	31.6 ± 1.1	ns
TAc (g/L tartaric acid)	5.90 ± 0.00	5.73 ± 0.06	5.63 ± 0.12	5.63 ± 0.06	ns	ns	ns	5.40 ± 0.10	ns
VA (g/L acetic acid)	0.32 ± 0.02	0.33 ± 0.02	0.56 ± 0.02 b	0.66 ± 0.02 a	***	***	**	0.94 ± 0.04 a	***
pH	3.48 ± 0.02 d	3.56 ± 0.02 c	3.65 ± 0.02 b	3.70 ± 0.01 a	***	***	ns	3.78 ± 0.02 a	**
TP (mg/L (+)-Catechin)	2270 ± 177.0 ab	2089 ± 54.5 b	2363 ± 127.5 a	2171 ± 164.9 ab	*	ns	ns	2446 ± 194.0	ns
TAns	991.5 ± 2.0	1053 ± 22.5	832.5 ± 73.2	846.7 ± 101.2	ns	ns	ns	801.0 ± 20.3	ns

^1^ Two-way ANOVA was used to compare data. Means followed by a different letter in a row are significant at *p*≤ 0.05 (Fisher’s LSD). All the stated uncertainty is a standard deviation of three replicates per treatment. ^2^ The asterisks indicate the level of significance of the two-way ANOVA for T, treatment, V, vintage, and T × V, interaction treatment × vintage: * *p* ≤ 0.05, ** *p* ≤ 0.01 and *** *p* ≤ 0.001, whereas ns indicates no significant differences. ^3^ Column B-C_sAF is compared with column B-C (B-C_iAF) for vintage 2021. ^4^ The olfactory thresholds (OTs) for the compounds from IBMP to ethyl acetate—the reference for each value was listed previously [54] or taken elsewhere [55,56]. The reported OTs for IBMP were 6 ng/L in the model solution (ethanol at 12% volume, 5 g/L tartaric acid, pH 3.5) and 15 ng/L in Merlot wine [57]. ^5^ nd—not detected.

**Table 3 foods-14-01101-t003:** Effects of the location (M, B) at two crop loads (C—control, CT—cluster thinning) on the Merlot wine volatiles and other compounds in two consecutive vintages (2020, 2021).

Crop Load	Significant Differences Between Locations B and M	
**Control (C)**	*Parameters with significantly higher values/concentrations at p ≤ 0.05* ^1^	*Parameters with significantly lower values/concentrations at p ≤ 0.05*
(1) Location B vs. Location M (2020)	**Ethanol**, ^2,4^ TAc, **IBMP**, **3MH**, **Ethyl decanoate**, **SUM acetates**, Benzyl alcohol, 2-Methyl-1-propanol, **2-Methyl-1-butanol**, **3-Methyl-1-butanol**, **2-Phenylethanol**, **SUM alcohols**	pH, Ethyl dodecanoate, **1-Hexanol**, **Ethyl acetate**
(2) Location B vs. Location M (2021)	Ethanol, ***VS***, ^3^ pH, ***TP***, ***3MH***, Ethyl dodecanoate, ***Ethyl palmitate***, 1-Butanol, ***γ-Butyrolactone***, ***Ethyl acetate***	***Hexyl acetate***, Isoamyl acetate, (Z)-3-hexen-1-ol
	**Significant Differences Between Locations B and M**	
**Cluster thinning (CT)**	*Parameters with significantly higher values/concentrations at p ≤ 0.05*	*Parameters with significantly lower values/concentrations at p ≤ 0.05*
(3) Location B vs. Location M (2020)	**Ethanol**, TDM, TP, **IBMP**, **3MH**, **Ethyl decanoate**, Ethyl octanoate, **SUM acetates**, Isoamyl acetate, 1-Butanol, **2-Methyl-1-butanol**, **3-Methyl-1-butanol**, **2-Phenylethanol**, **SUM alcohols**	TAc, Ethyl palmitate, **1-Hexanol**, 2-Methyl-1-propanol, **Ethyl acetate**
(4) Location B vs. Location M (2021)	***VA***, ***TP***, ***3MH***, Ethyl butyrate, Ethyl octanoate, Ethyl hexanoate, ***Ethyl palmitate***, SUM EEFAs, 2-Phenethyl acetate, SUM acetates, Benzyl alcohol, 3-Methyl-1-butanol, SUM alcohols, Diethyl succinate, ***γ-Butyrolactone***, ***Ethyl acetate***	IBMP, ***Hexyl acetate***, 1-Hexanol

^1^ Student *t*-test was used to compare data. Values for each parameter at different locations (M, B), crop loads (C, CT), and vintages (2020, 2021) are shown in Table 1 (location M) and Table 2 (location B). ^2^ The parameters in bold, which differ significantly, followed the same trend when comparing the two locations in 2020, regardless of the crop load (first and third rows in the table). ^3^ The parameters in bold and italic, which differ significantly, followed the same trend when comparing the two locations in 2021, regardless of the crop load (second and fourth rows in the table). ^4^ The underlined parameters followed the same trend when the two locations were compared at C (first and second rows in the table) or CT crop load (third and fourth rows in the table), regardless of the vintage.

**Table 4 foods-14-01101-t004:** Triangle tests for the Merlot wines from two locations (M and B) regarding two crop loads (cluster thinning (CT) vs. control (C)) and two types of fermentations (iAF vs. sAF) in two vintages (2020, 2021) (*n* = 8 responses).

LOCATION M	Comparisons of Two Yields and Two Alcoholic Fermentations			
	M-CT vs. M-C		M-C_iAF (M-C) vs. M-C_sAF	
Vintage	2020	2021	2020	2021
Triangle test	ns ^1^	ns	Significant at *a* < 0.01	Significant at *a* < 0.001
**LOCATION B**	**Comparisons of Two Yields and Two Alcoholic Fermentations**			
	**B-CT vs. B-C**		**B-C_iAF (B-C) vs. B-C_sAF**	
Vintage	2020	2021	2020	2021
Triangle test	ns	ns	/ ^2^	Significant at *a* < 0.01
**LOCATIONS**	**Comparisons of Two Locations**			
	**LOCATION B vs. LOCATION M (C)**		**LOCATION B vs. LOCATION M (CT)**	
Vintage	2020	2021	2020	2021
Triangle test	Significant at *a* < 0.001	Significant at *a* < 0.001	Significant at *a* < 0.01	Significant at *a* < 0.001

^1^ ns indicates no significant differences. ^2^ Spontaneous alcoholic fermentation was not performed.

## Data Availability

The original data presented in this study are openly available in FigShare at https://doi.org/10.6084/m9.figshare.28547390.

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
