# Peer review of "Influence of Vineyard Location, Cluster Thinning and Spontaneous Alcoholic Fermentation on Wine Composition"

_foods, 2025, doi:10.3390/foods14071101_

Round 1
Reviewer 1 Report
Comments and Suggestions for Authors
The article describes the effect of several factors on the chemical composition and sensory features of Merlot wines produced in two different locations. The article is well written, and the results are interesting, mainly from a technical point of view. However, there are a series of questions that make it unsuitable to be published as it is.
Main questions
- Quality concepts
The authors describe quality, but they do not refer to which concept of quality. The issue of fine wine quality is not addressed; that is, aesthetic features like complexity were not included, and while balance and persistence were evaluated, their scores were not reported. Indeed, what authors describe is the commercial quality, defined as the whole of features that are pleasant to most consumers. Although there were 8 professional tasters, their responses were consistent with this sort of quality. In this case, quality could not be distinguished from preference or liking. The most preferred wine was from B location, characterized by darker colour and higher ethanol content, exactly consistent with the features of the red wines awarded gold medals in international wine challenges. This wine was most likely more intense, fruity and smooth, but these features were not reported. By a top-down effect, the higher colour should have conditioned all other descriptors since wines were not tasted in dark glasses. There are references in literature that describe these issues, and authors should include it in a limitations section.
As a result, authors should replace “better” with “most liked” or “most appreciated”.
In addition, the wines were too young (6-9 months), even Merlot, to be assessed as red wines ready for consumption.
- Wine comparisons
Given that the control wines were different, it makes no sense to compare the factors between the two wines. That is, since one wine was considered better, any effect may always result in better appreciation simply because the wine appreciation was high enough to accommodate any differences (see Table 5). Therefore, these comparisons should be remove,d and the authors should explain why.
The higher liking may be explained by the origin of B wine, from silty-loamy soils, while M wine came from sandy-loamy soils, which may be responsible for producing bolder styles. In addition, the training systems were different. These bolder styles are also more frequent when the climate is also warmer and less rainy, as in B location. In addition, the climate might have led to higher rot in M location, but the health status was not described. The microbial load before fermentation is consistent with rot since healthy berries usually bear much less than 4 log CFU/g. The absence of acetic bacteria or gluconic acid determination could have clarified this issue, as widely described in literature.
- Microbial identifications
The genus of yeasts are reported as Saccharomyces, Hanseniaspora, Starmerella or others, but the identification was based on cellular morphological observation. In some cases, this may be easy; in other cases, it is not precise, mainly when dealing with a scientific paper.
By the way, italicize all mentions of microbial genuses.
Secondary questions
- Paper length
The paper should be shortened by half without losing information in the introduction and discussion.
Rephrase the abstract (the first sentence is not fit to an abstract) and the conclusions (finishing with bullets is not usual).
- Anova and Multivariate analysis
Reduce the noise by removing those parameters that are different among the trials (mentioned as ns) and use clear figures (not dark) colour.
Remove colour from the tables.
Oddly, acetaldehyde is 22 mg/l in one case, but it is not different from the concentrations that are more than half this value.
- Sensory analysis
The triangle test was reported as n=24, but there were only 8 assessors. The 3 sessions should not be merged since the differences are only between the wines tasted in one session. At a 5% significance level, the correct answers should be 6 out of 8, while it should be 13 out of 24. So, there are probably no differences in all trials.
In conclusion, if the authors address these issues, we will happily review the article again.
Author Response
Main questions
- Quality concepts
Comment 1: The authors describe quality, but they do not refer to which concept of quality. The issue of fine wine quality is not addressed; that is, aesthetic features like complexity were not included, and while balance and persistence were evaluated, their scores were not reported. Indeed, what authors describe is the commercial quality, defined as the whole of features that are pleasant to most consumers. Although there were 8 professional tasters, their responses were consistent with this sort of quality. In this case, quality could not be distinguished from preference or liking. The most preferred wine was from B location, characterized by darker colour and higher ethanol content, exactly consistent with the features of the red wines awarded gold medals in international wine challenges. This wine was most likely more intense, fruity and smooth, but these features were not reported. By a top-down effect, the higher colour should have conditioned all other descriptors since wines were not tasted in dark glasses. There are references in literature that describe these issues, and authors should include it in a limitations section.
As a result, authors should replace “better” with “most liked” or “most appreciated”.
In addition, the wines were too young (6-9 months), even Merlot, to be assessed as red wines ready for consumption.
Response 1: In light of this comment and comment 10 (point 3 in secondary questions), we have removed the two-sided directional difference test (2-SDD) from the Materials and Methods (lines 399-404, 408-411; text highlighted in yellow) and Results (lines 736-772; text highlighted in yellow, consequently paragraph 3.5 in the Results now has no sub-chapters) and Discussion (lines 777-906; text highlighted in yellow) to increase relevance. As a result, we have removed some of the text from the Introduction (lines 120-135) and corrected Tables 4 and 5 and merged them into a new Table 4. In addition, in light of the comment above, we have replaced the term wine quality with wine composition and the term wine sensory quality with the term wine sensory characteristics throughout the text (highlighted in yellow). We agree with the reviewer's comment regarding the influence of one attribute on another due to the lack of use of black glasses, and this was also one of the reasons for the retraction of the 2-SDD test from the entire manuscript.
It is true that 6-9 months is quite early to assess the sensory quality of red wine, but in view of the changes described above, we believe that this remark is no longer relevant.
- Wine comparisons
Comment 2: Given that the control wines were different, it makes no sense to compare the factors between the two wines. That is, since one wine was considered better, any effect may always result in better appreciation simply because the wine appreciation was high enough to accommodate any differences (see Table 5). Therefore, these comparisons should be removed and the authors should explain why.
Response 2: We explained in the answer to point 1 of the main questions (Response 1).
Comment 3: The higher liking may be explained by the origin of B wine, from silty-loamy soils, while M wine came from sandy-loamy soils, which may be responsible for producing bolder styles. In addition, the training systems were different. These bolder styles are also more frequent when the climate is also warmer and less rainy, as in B location. In addition, the climate might have led to higher rot in M location, but the health status was not described. The microbial load before fermentation is consistent with rot since healthy berries usually bear much less than 4 log CFU/g. The absence of acetic bacteria or gluconic acid determination could have clarified this issue, as widely described in literature.
Response 3: In view of the withdrawal of the 2-SDD test, this remark is largely obsolete. The health status of the grapes was healthy, undamaged grapes (added in the text in line 223). Since some time elapses between de-stemming and crushing and filling the fermentors, the yeast population also increases during this time and is higher than in the grapes themselves. There are also visible differences in the number of microorganisms in the initial phase of AF between vintages and locations.
- Microbial identifications
Comment 4: The genus of yeasts are reported as Saccharomyces, Hanseniaspora, Starmerella or others, but the identification was based on cellular morphological observation. In some cases, this may be easy; in other cases, it is not precise, mainly when dealing with a scientific paper.
Response 4: The identification of yeast strains was not the main objective of our study. The yeast population section was added to ensure that we were sure that there were indeed differences in the population between inoculated and spontaneous AF. We believe that the approach we had for monitoring the microbiological population is sufficient for this purpose. We also have extensive experience in monitoring yeast populations during winemaking, as you can see from the references of our previous studies (references 14 and 15).
By the way, italicize all mentions of microbial genus.
Corrected throughout the text (lines 485-528).
Secondary questions
- Paper length
Comment 5: The paper should be shortened by half without losing information in the introduction and discussion.
Response 5: The paper has been considerably shortened in all paragraphs by the changes mentioned above and below. However, it is true that the paper describes the influence of three factors on the composition of wine, so we certainly cannot shorten the paper by half, as this would lose a lot of relevant information for the field of research. The journal also does not limit the number of pages in research papers.
Comment 6: Rephrase the abstract (the first sentence is not fit to an abstract) and the conclusions (finishing with bullets is not usual).
Response 6: We have corrected and added. The changes are marked in yellow bold (lines 8-9 and 954-962).
- Anova and Multivariate analysis
Comment 7: Reduce the noise by removing those parameters that are different among the trials (mentioned as ns) and use clear figures (not dark) colour.
Response 7: In the text of the results, we have only indicated and highlighted those results that are significantly different. We believe that the other results in Tables 1 and 2, including ns, are also meaningful, especially with regard to comparing the content of compounds between different studies (using the data for meta-studies). It would also be difficult to find an overlap between compounds that could be removed in all statistical comparisons in Tables 1-3. We have changed the background color of the Figures 1b-3b as suggested.
Comment 8: Remove colour from the tables.
Response 8: We have deleted the color highlighting of text in tables and corrected the table legends accordingly.
Comment 9: Oddly, acetaldehyde is 22 mg/l in one case, but it is not different from the concentrations that are more than half this value.
Response 9: True. But as mentioned in MM, for compounds with inhomogeneous variances, we had to use the Kruskall-Wallis test, which is much more stringent than the LSD test and consequently requires larger differences between means for significant differences. Acetaldehyde is one of these compounds.
- Sensory analysis
Comment 10: The triangle test was reported as n=24, but there were only 8 assessors. The 3 sessions should not be merged since the differences are only between the wines tasted in one session. At a 5% significance level, the correct answers should be 6 out of 8, while it should be 13 out of 24. So, there are probably no differences in all trials.
Response 10: We took the reviewer's comment into account and considered only one repetition of the assessment for the triangle test and corrected the number of n to 8 (n=8) throughout the text (highlighted in yellow, lines 408, 763 and 775). As a result, the probability with which we confirmed statistical differences between treatments changed, which we have corrected and highlighted in Tables 4 and 5 and now in the combined Table 4. As the number of panellists is now too small for the 2-SDD test, the results of this test are not relevant and have been removed from the paper.
Reviewer 2 Report
Comments and Suggestions for Authors
The manuscript is clearly written, what makes reading an easy task. The article presents a well-structured study on the impact of vineyard location, cluster thinning, and spontaneous alcoholic fermentation on Merlot wine quality.
The introduction is clear and focus on the main parameters that will be studied. The study was conducted over two years (2020 and 2021) at two vineyard locations (B and M) in Slovenia.
The Materials and methods section are too long. Consider the possibility of summarizing and moving some text to the Results section.
In Material and methods, line 180 and 190 it is referred Figure 1, but should be Figure S1, right?
The text describing the figure (lines 179-198) are results? Should be moved to Results section.
In line 234 it is referred Figure 3, but should be Figure S3.
The vinification of grapes was performed in 15 L volume in the case of inoculated alcoholic fermentation (iAF) while for spontaneous alcoholic fermentation (sAF) a volume of only 3L was used. The use of smaller fermenters for sAF vs. 15 L for iAF may affect fermentation dynamics and heat dissipation. Please clarify and explain how the different fermenter sizes (3 L vs. 15 L) impact fermentation conditions.
In line 242 it is mentioned the bottling of the wines. Please complete information on the number of bottles.
In lines 245-248 complete the information on the number of bottles that were used for sensory analysis.
No details on replicates per sample for chemical analysis—were all tests performed in triplicate? Please clarify how was it done for all samples.
In Results section, the most consistent and significant effects obtained should be emphasized and practical relevance highlighted, not only reading the statistical data.
Correct to italic, genera/species names of microorganisms.
The comparison of locations M and B discussed in line 609 and onward, including table 3, refer to data obtained from iAF? Please clarify.
The size of the Discussion section could be significantly reduced, making the discussion more concise and structured.
In the conclusions, it is not clear whether analytical differences resulted in perceptible sensory changes in the wine. Can you elaborate on that?
Author Response
Comment 1: The Materials and methods section are too long. Consider the possibility of summarizing and moving some text to the Results section.
Response 1: We have moved part of the text from the Materials and methods to the Results – see below.
Comment 2: In Material and methods, line 180 and 190 it is referred Figure 1, but should be Figure S1, right?
Response 2: That is correct; corrected and thanks for the notice.
Comment 3: The text describing the figure (lines 179-198) are results? Should be moved to Results section.
Response 3: We have moved this part of the text (lines 179 to 198) to the results (lines 429-448).
Comment 4: In line 234 it is referred Figure 3, but should be Figure S3.
Response 4: Instead of Figure 3, the correct text is Table S5 – corrected (line 237).
Comment 5: The vinification of grapes was performed in 15 L volume in the case of inoculated alcoholic fermentation (iAF) while for spontaneous alcoholic fermentation (sAF) a volume of only 3L was used. The use of smaller fermenters for sAF vs. 15 L for iAF may affect fermentation dynamics and heat dissipation. Please clarify and explain how the different fermenter sizes (3 L vs. 15 L) impact fermentation conditions.
Response 5: It is true, we agree that the difference in fermentor volume can influence the dynamics of the yeast population. However, we believe that in both cases these are microvinifications and that possible differences in T due to the course of AF did not have a decisive influence on the concentration of yeast cells and the representation of species during AF. Now, we have not included this explanation in the text of the paper. If the reviewer deems it necessary, we can add it.
Comments 6: In line 242 it is mentioned the bottling of the wines. Please complete information on the number of bottles.
In lines 245-248 complete the information on the number of bottles that were used for sensory analysis.
No details on replicates per sample for chemical analysis—were all tests performed in triplicate? Please clarify how was it done for all samples.
Response 6: We have clarified all three ambiguities mentioned above by adding information on the number of bottles and repeated measurements within the text in lines 249-264.
Comment 7: In Results section, the most consistent and significant effects obtained should be emphasized and practical relevance highlighted, not only reading the statistical data.
Response 7: In the results, we have mentioned all statistically significant and consistent differences between the treatments (example text in lines 531-537). As the length of the paper is already considerable and one of the reviewers also suggested shortening the paper by half, we have not further explained the differences in results between treatments that we consider less relevant. However, we have commented on the possible reasons for the differences in the measured parameters in the discussion and not in the results.
Comment 8: Correct to italic, genera/species names of microorganisms.
Response 8: We have corrected the text.
Comment 9: The comparison of locations M and B discussed in line 609 and onward, including table 3, refer to data obtained from iAF? Please clarify.
Response 9: It is true; we have explained this fact in more detail in lines 668-669.
Comment 10: The size of the Discussion section could be significantly reduced, making the discussion more concise and structured.
Response 10: As we have removed some of the sensory analysis in all paragraphs at the suggestion of another reviewer, the discussion paragraph has also been shortened. However, we believe it is important to link our findings as relevantly as possible to the results of other similar studies on the factors examined.
Comment 11: In the conclusions, it is not clear whether analytical differences resulted in perceptible sensory changes in the wine. Can you elaborate on that?
Response 11: We have added a conclusion (iii) in which we explain the above-mentioned connection (lines 958-960).
Reviewer 3 Report
Comments and Suggestions for Authors
In this work, the influence of three factors (vineyard location -2-, crop load per vine, and type of alcoholic fermentation –inoculated and spontaneous-) on the quality of Merlot wine during two consecutive vintages (2020 and 2021) was investigated. The wine quality were analyzed by using a metabolomic approach, and by microbiological, chemical and sensory analyses.
The basic idea of this study is of interest for the production of Merlot wines in this country in order to improve their quality. The paper is well organized and presented, but several mistakes and explanations need to be included:
Line 38: IBMP: The first time an abbreviation is quoted, it must be explained.
Line 80: Starmerella (not Starmelerra).
Line 223: mg/L (not mg/l).
Line 239: “measured”: the malic acid and lactic acid in the wines were then determined.
Line 275: 2.4.2. Analysis of must and wine parameters (not “2.4.2. Analysis of standard must and wine parameters”).
Line 276: “Total soluble solids (TSS) in the must were determined with…). Analytical parameters are determined by measuring some property, for example, here, the refractive index.
Line 282: The ethanol content of the 281 wine was determined… (not measured).
Line 283: the density of the wine was determined… (not measured).
Line 285: The total dry matter (TDM) of the wine was determined… (not measured).
Line 288: Tartaric acid, malic acid, lactic acid, and citric acid were determined (or quantified)… (not measured).
Line 294: Please give a brief description of the chromatographic conditions.
Lines 301 and 302: mg/L (not mg L-1.).
Line 307: 50 mL.
Line 308: 4MMP: The first time an abbreviation is quoted, it must be explained.
Line 318: Please give a brief description of the chromatographic conditions.
Line 327: Volume and concentration of the internal standard 4-methyl-2-pentanol?
Line 331: No internal standards were added for the analysis of these compounds? If yes, which one (volume and concentration).
Line 338: Please indicate the identification and quantification of the compounds. References such as 50 are not easily accessible.
Lines 341 and 347: Include a space between numbers and unit (ºC).
Line 347: 30 mL.
Lines 382, 383 and all manuscript and tables: The statistical “p” in italics.
Tables S1 and S2: Lorenz et al., 1994 (not Lorenz et.at, 1994).
Table 1: 1-hexanol, Benzyl alcohol and (Z)-3-hexen-1-ol are NOT higher alcohols. They are alcohols; therefore, the authors should write ‘sum of alcohols’ (not higher alcohols) or separate these 3 compounds from the sum of higher alcohols. As they would have to redo all the statistics, I recommend that, throughout the manuscript, they remove the adjective ‘higher’.
The statistical analysis has been exhaustive.
Author Response
Comments 1:
Line 38: IBMP: The first time an abbreviation is quoted, it must be explained.
Line 80: Starmerella (not Starmelerra).
Line 223: mg/L (not mg/l).
Line 239: “measured”: the malic acid and lactic acid in the wines were then determined.
Line 275: 2.4.2. Analysis of must and wine parameters (not “2.4.2. Analysis of standard must and wine parameters”).
Line 276: “Total soluble solids (TSS) in the must were determined with…). Analytical parameters are determined by measuring some property, for example, here, the refractive index.
Line 282: The ethanol content of the 281 wine was determined… (not measured).
Line 283: the density of the wine was determined… (not measured).
Line 285: The total dry matter (TDM) of the wine was determined… (not measured).
Line 288: Tartaric acid, malic acid, lactic acid, and citric acid were determined (or quantified)… (not measured).
Line 294: Please give a brief description of the chromatographic conditions.
Lines 301 and 302: mg/L (not mg L-1.).
Line 307: 50 mL.
Line 308: 4MMP: The first time an abbreviation is quoted, it must be explained.
Line 318: Please give a brief description of the chromatographic conditions.
Line 327: Volume and concentration of the internal standard 4-methyl-2-pentanol?
Line 331: No internal standards were added for the analysis of these compounds? If yes, which one (volume and concentration).
Line 338: Please indicate the identification and quantification of the compounds. References such as 50 are not easily accessible.
Lines 341 and 347: Include a space between numbers and unit (ºC).
Line 347: 30 mL.
Lines 382, 383 and all manuscript and tables: The statistical “p” in italics.
Tables S1 and S2: Lorenz et al., 1994 (not Lorenz et.at, 1994).
Response 1: We have entered all the proposed changes in the line numbers mentioned or slightly shifted due to additional text entries and marked them in yellow.
Lines 298-301 and 343-251 now list the chromatographic conditions for the analysis of the organic acids and varietal thiols.
The concentrations of the internal standards (lines 359-368) and details on the identification and quantification of the compounds from section 2.4.8 have also been added.
Comment 2: Table 1: 1-hexanol, Benzyl alcohol and (Z)-3-hexen-1-ol are NOT higher alcohols. They are alcohols; therefore, the authors should write ‘sum of alcohols’ (not higher alcohols) or separate these 3 compounds from the sum of higher alcohols. As they would have to redo all the statistics, I recommend that, throughout the manuscript, they remove the adjective ‘higher’.
Response 2: We have taken the reviewer's comment into account and corrected the naming of the sum of alcohols in Tables 1-3 and all three Graphs 1b-3b. We have done the same for the acetates. We have therefore corrected the naming of HAA and HA throughout the text and marked them in yellow bold.
Comment 3: The statistical analysis has been exhaustive.
Response 3: We agree, but this is necessary given the statistical design of the study.
Round 2
Reviewer 1 Report
Comments and Suggestions for Authors
The authors have answered questions.
Only one additional issue:
1. Explain that species identification are presumptive because were only done by morphologiacl observation
Author Response
Comment 1: Explain that species identification are presumptive because were only done by morphological observation.
Response 1: In lines 444-447 (text highlighted in yellow), we now explain the limitation of the methodology we use to follow the dynamics of the yeast population during alcoholic fermentation.